# Ancient viral genomes reveal introduction of human pathogenic viruses into Mexico during the transatlantic slave trade

Axel A Guzmán-Solís[1], Viridiana Villa-Islas[1], Miriam J Bravo-López[1], Marcela Sandoval-Velasco[2], Julie K Wesp[3], Jorge A Gómez-Valdés[4], María de la Luz Moreno-Cabrera[5], Alejandro Meraz[5], Gabriela Solís-Pichardo[6], Peter Schaaf[7], Benjamin R TenOever[8], Daniel Blanco-Melo[8,9]*, María C Ávila Arcos[1]*

[1]Laboratorio Internacional de Investigación sobre el Genoma Humano, Universidad Nacional Autónoma de México, Querétaro, Mexico; [2]Section for Evolutionary Genomics, The Globe Institute, Faculty of Health, University of Copenhagen, Copenhagen, Denmark; [3]Department of Sociology and Anthropology, North Carolina State University, Raleigh, United States; [4]Escuela Nacional de Antropología e Historia, Mexico City, Mexico; [5]Instituto Nacional de Antropología e Historia, Mexico City, Mexico; [6]Laboratorio Universitario de Geoquímica Isotópica (LUGIS), Instituto de Geología, Universidad Nacional Autónoma de México, Mexico City, Mexico; [7]LUGIS, Instituto de Geofísica, Universidad Nacional Autónoma de México, Mexico City, Mexico; [8]Department of Microbiology, Icahn School of Medicine at Mount Sinai, New York, United States; [9]Vaccine and Infectious Disease Division, Fred Hutchinson Cancer Research Center, Seattle, WA, United States

*For correspondence:
dblancom@fredhutch.org (DB-M);
mavila@liigh.unam.mx (MCAA)

Competing interest: The authors declare that no competing interests exist.

**ABSTRACT** After the European colonization of the Americas, there was a dramatic population collapse of the Indigenous inhabitants caused in part by the introduction of new pathogens. Although there is much speculation on the etiology of the Colonial epidemics, direct evidence for the presence of specific viruses during the Colonial era is lacking. To uncover the diversity of viral pathogens during this period, we designed an enrichment assay targeting ancient DNA (aDNA) from viruses of clinical importance and applied it to DNA extracts from individuals found in a Colonial hospital and a Colonial chapel (16th–18th century) where records suggest that victims of epidemics were buried during important outbreaks in Mexico City. This allowed us to reconstruct three ancient human parvovirus B19 genomes and one ancient human hepatitis B virus genome from distinct individuals. The viral genomes are similar to African strains, consistent with the inferred morphological and genetic African ancestry of the hosts as well as with the isotopic analysis of the human remains, suggesting an origin on the African continent. This study provides direct molecular evidence of ancient viruses being transported to the Americas during the transatlantic slave trade and their subsequent introduction to New Spain. Altogether, our observations enrich the discussion about the etiology of infectious diseases during the Colonial period in Mexico.

## Introduction

European colonization in the Americas resulted in a frequent genetic exchange mainly between Native American populations, Europeans, and Africans (*Aguirre-Beltrán, 2005*; *Rotimi et al., 2016*; *Salas et al., 2004*). Along with human migrations, numerous new species were introduced to the Americas including bacterial and viral pathogens, which played a major role in the dramatic population collapse

**eLife digest** The arrival of European colonists to the Americas, beginning in the 15[th] century, contributed to the spread of new viruses amongst Indigenous people. This led to massive outbreaks of disease, and millions of deaths that caused an important Native population to collapse. The exact viruses that caused these outbreaks are unknown, but smallpox, measles, and mumps are all suspected.

During these times, traders and colonists forcibly enslaved and displaced millions of people mainly from the West Coast of Africa to the Americas. The cruel, unsanitary, and overcrowded conditions on ships transporting these people across the Atlantic contributed to the spread of infectious diseases onboard. Once on land, infectious diseases spread quickly, partly due to the poor conditions that enslaved and ndigenous people were made to endure. Native people were also immunologically naïve to the newly introduced pathogens, making them susceptible to severe or fatal outcomes. The new field of paleovirology may help scientists identify the viruses that were circulating in the first years of colonization and trace how viruses arrived in the Americas.

Using next-generation DNA sequencing and other cutting-edge techniques, Guzmán-Solís et al. extracted and enriched viral DNA from skeletal remains dating back to the 16[th] century. These remains were found in mass graves that were used to bury epidemic victims at a colonial hospital and chapel in what is now Mexico City. The experiments identified two viruses, human parvovirus B19 and a human hepatitis B virus. These viral genomes were recovered from human remains of first-generation African people in Mexico, as well as an individual who was an Indigenous person.

Although the genetic material of these ancient viruses resembled pathogens that originated in Africa, the study did not determine if the victims died from these viruses or another cause. On the other hand, the results indicate that viruses frequently found in modern Africa were circulating in the Americas during the slave trade period of Mexico. Finally, the results provide evidence that colonists who forcibly brought African people to the Americas participated in the introduction of viruses to Mexico. This constant influx of viruses from the old world, led to dramatic declines in the populations of Indigenous people in the Americas.

that afflicted the immunologically naïve Indigenous inhabitants (*Acuña-Soto et al., 2004*; *Lindo et al., 2016*). Among these pathogens, viral diseases, such as smallpox, measles, and mumps, have been proposed to be responsible for many of the devastating epidemics during the Colonial period (*Acuña-Soto et al., 2004*). Remarkably, the pathogen(s) responsible for the deadliest epidemics reported in New Spain (the Spanish viceroyalty that corresponds to Mexico, Central America, and the current US southwest states) remains unknown and is thought to have caused millions of deaths during the 16th century (*Acuña-Soto et al., 2004*). Indigenous populations were drastically affected by these mysterious epidemics, generically referred to as *Cocoliztli* ('pest' in Nahuatl), followed by Africans and to a lesser extent European people (*Acuña-Soto et al., 2004*; *Malvido and Viesca, 1982*; *Somolinos d'Árdois, 1982*). Accounts of the 1576 *Cocoliztli* epidemic were described in autopsy reports of victims treated at the 'Hospital Real de San José de los Naturales' (HSJN) (*Malvido and Viesca, 1982*; *Wesp, 2017*), the first hospital in Mexico dedicated specifically to treat the Indigenous population (*Malvido and Viesca, 1982*; *Wesp, 2017*; *Figure 1a and b*).

The study of ancient viral genomes has revealed important insights into the evolution of specific viral families (*Barquera et al., 2020*; *Duggan et al., 2016*; *Düx et al., 2020*; *Kahila Bar-Gal et al., 2012*; *Krause-Kyora et al., 2018*; *Mühlemann et al., 2018a*; *Mühlemann et al., 2018a*; *Mühlemann et al., 2018b*; *Neukamm et al., 2020*; *Pajer et al., 2017*; *Patterson Ross et al., 2018*; *Xiao et al., 2013*), as well as their interaction with human populations (*Spyrou et al., 2019*). To explore the presence of viral pathogens in circulation during epidemic periods in New Spain, we leveraged the vast historical and archeological information available for the Colonial HSJN. These include the skeletal remains of over 600 individuals recovered from mass burials associated with the hospital's architectural remnants (*Figure 1b*). Many of these remains were retrieved from burial contexts suggestive of an urgent and simultaneous disposal of the bodies, as in the case of an epidemic (*Meza, 2013*; *Wesp, 2017*). Prior bioarcheological research has shown that the remains of a number of individuals in the HSJN collection displayed dental modifications and/or morphological indicators typical of African

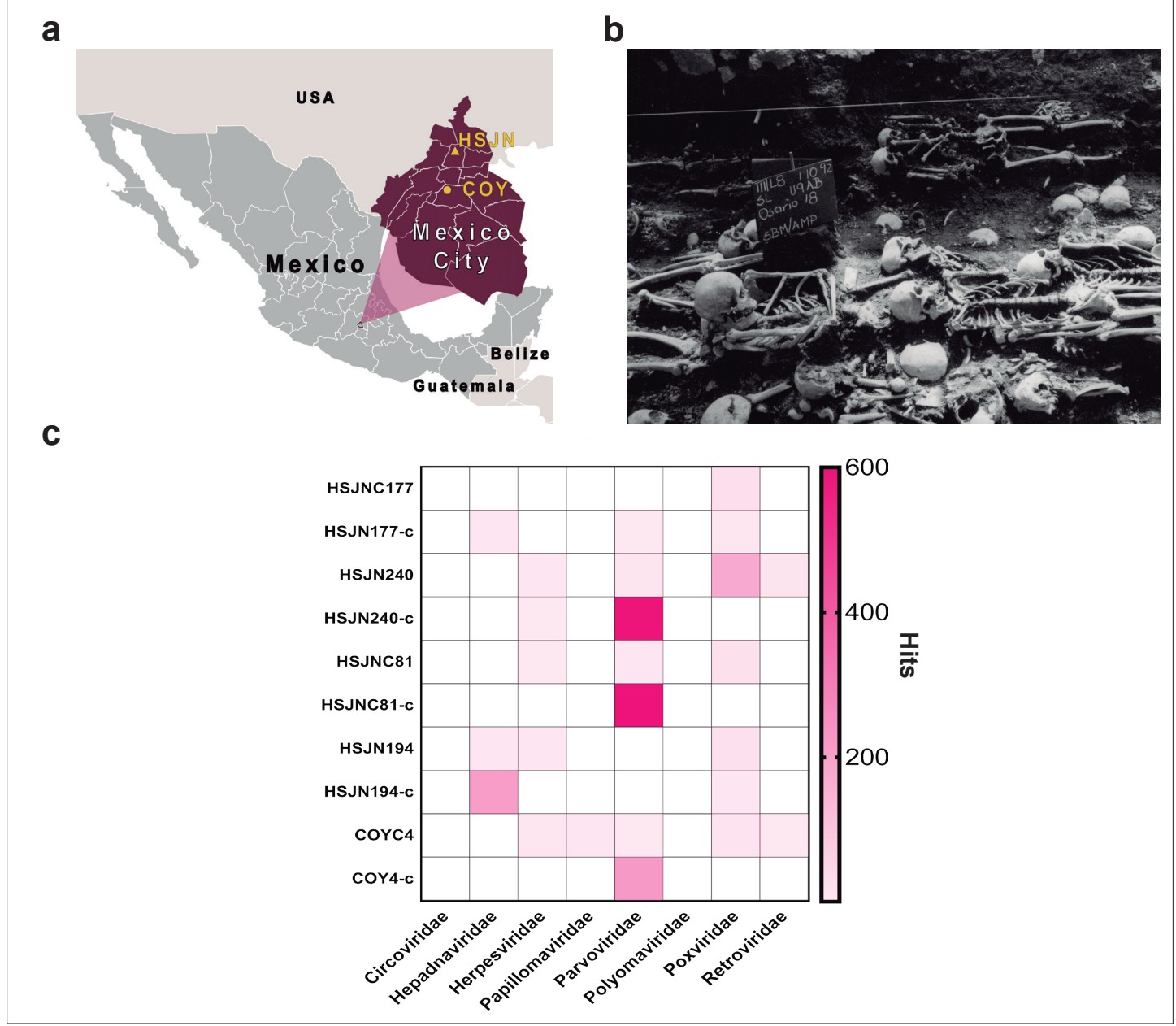

**Figure 1.** Metagenomic analysis of Colonial individuals reveal HBV-like and B19V-like hits. (**a**) Location of the archeological sites used in this study, HSJN (19.431704–99.141740) is shown as a yellow triangle and COY (19.347079–99.159017) as a yellow circle, lines in pink map show current division of Mexico City. (**b**) Several individuals discovered in mass burials archaeologically associated with the Hospital Real de San José de los Naturales (HSJN) and Colonial epidemics. (**c**) Metagenomic analysis performed with MALT 0.4.0 based on the Viral NCBI RefSeq. Viral abundances were compared and normalized automatically in MEGAN between shotgun (*sample_name*) and capture (*sample_name*-c) next-generation sequencing (NGS) data. Only HBV or B19V-positive samples are shown (all samples analyzed are shown in *Figure 1—figure supplements 2–3*). A capture negative control (HSJN177) is shown.

© 2016, Secretaria de Cultura INAH, SINAFO, Fototeca DSA. Panel b was taken by Salvador Pulido Méndez and is reproduced from the "Proyecto Metro Línea 8" with permission from "Secretaria de Cultura INAH, SINAFO, Fototeca DSA". This panel is not covered by the CC-BY 4.0 licence and further reproduction of this panel would need permission from the copyright holder.

The online version of this article includes the following figure supplement(s) for figure 1:

**Figure supplement 1.** Pipeline followed for ancient viral genomes reconstruction.

**Figure supplement 2.** Individuals with DNA traces of clinical important viral families.

**Figure supplement 3.** Target viral abundances post-capture.

*Figure 1 continued on next page*

**Figure supplement 7.** Bacterial metagenomic analysis from COYC4.

ancestry (*Meza, 2013*), consistent with historical and archeological research that documents the presence of a large number of both free and enslaved Africans and their descendants in Colonial Mexico (*Aguirre-Beltrán, 2005*). Indeed, a recent paleogenomics study reported a sub-Saharan African origin of three individuals from this collection (*Barquera et al., 2020*).

Here we describe the recovery and characterization of viral pathogens that circulated in New Spain during the Colonial period, using ancient DNA (aDNA) techniques (*Figure 1—figure supplement 1*). For this work, we sampled skeletal human remains recovered from the HSJN where archeological context suggest victims of epidemics were buried (*Meza, 2013*) and from 'La Concepcion' chapel, one of the first catholic conversion centers in New Spain (*Moreno-Cabrera et al., 2015*; *Figure 1a*). We report the reconstruction of ancient hepatitis B virus (HBV) and human parvovirus B19 (B19V) genomes recovered from these remains, providing a direct molecular evidence of human viral pathogens of African origin being introduced to New Spain during the transatlantic slave trade.

## Results

We sampled the skeletal remains from two archeological sites, a Colonial Hospital and a Colonial chapel in Mexico City (*Figure 1a and b*). For the HSJN, 21 dental samples (premolar and molar teeth) were selected based on previous morphometric analyses and dental modifications that suggested an African ancestry (*Hernández-Lopez and Negrete, 2012*; *Karam-Tapia, 2012*; *Meza, 2013*; *Ruíz-Albarrán, 2012*). The African presence in the Indigenous Hospital might reflect an urgent response to an epidemic outbreak since hospitals treated patients regardless of the origin of the affected individuals during serious public health crises (*Meza, 2013*). Dental samples of five additional individuals were selected (based on their conservation state) from 'La Concepción' chapel (COY), which is located 10 km south of the HSJN in Coyoacán, a Pre-Hispanic Indigenous neighborhood that became the first Spanish settlement in Mexico City after the fall of Tenochtitlan (*Moreno-Cabrera et al., 2015*). Following strict aDNA protocols, we processed these dental samples to isolate aDNA for next-generation sequencing (NGS) (*Figure 1—figure supplement 1*, Materials and methods). Tooth roots (which are vascularized) can be a good source of pathogen DNA (*Key et al., 2017*), especially in the case of viruses that are widespread in the bloodstream during systemic infection. Accordingly, a number of previous studies have successfully recovered ancient viral DNA from tooth roots (*Barquera et al., 2020*; *Krause-Kyora et al., 2018*; *Mühlemann et al., 2020*; *Mühlemann et al., 2018a*; *Mühlemann et al., 2018b*).

Metagenomic analyses with MALT (*Herbig et al., 2018*) (Materials and methods) on the NGS data using the Viral NCBI RefSeq database as a reference (*Pruitt et al., 2007*) revealed 17 samples containing at least one normalized hit to viral DNA (abundances were normalized to the smallest library since each sample had different number of reads) (Materials and methods), particularly similar to *Hepadnaviridae*, *Herpesviridae*, *Parvoviridae,* and *Poxviridae* viral families (*Figure 1c*, *Figure 1—figure supplement 2*, Materials and methods). These viral hits revealed the potential to recover ancient viral genomes from these samples. We selected 12 samples for further screening (*Figure 1c*, *Figure 1—figure supplement 3*) based on the DNA concentration of the NGS library and the quality of the hits to a clinically important virus (HBV, B19V, papillomavirus, smallpox).

To isolate and enrich the viral DNA fraction in the sequencing libraries, biotinylated single-stranded RNA probes designed to capture sequences from diverse human viral pathogens were synthesized (*Supplementary file 1A*). The selection of the viruses included in the capture design considered the following criteria: (1) DNA viruses previously retrieved from archeological human remains (i.e., hepatitis B virus, human parvovirus B19, variola virus), (2) representative viruses from families capable of integrating into the human genome (i.e., *Herpesviridae*, *Papillomaviridae*, *Polyomaviridae*, *Circoviridae*), or (3) RNA viruses with a DNA intermediate (i.e., *Retroviridae*). Given the size constraints of

the probe kit, only a couple of genes were selected from some viral families (Materials and methods, *Supplementary file 1A*). Additionally, a virus-negative aDNA library, which showed no hits to any viral family included in the capture assay (except for a frequent *Poxviridae*-like region identified as an Alu repeat; *Tithi et al., 2018*) was captured and sequenced as a negative control (HSJN177) to estimate the efficiency of our capture assay. Only one post-capture library had an ~100 -fold increase of *Hepadnaviridae*-like hits (HBV), while three more libraries had an ~50–200 -fold increase of *Parvoviridae*-like hits (B19V) (*Figure 1c*, *Supplementary file 1B*), compared to their corresponding pre-capture libraries (Materials and methods). In contrast, the captured negative control (HSJN177) presented a negligible enrichment of these viral hits (*Figure 1c*, *Supplementary file 1B*).

Independently, a metagenomic analysis using Kraken2 (*Wood et al., 2019*) and Pavian (*Breitwieser and Salzberg, 2020*) was performed on the non-human (unmapped) reads as part of a different study (*Bravo-Lopez et al., 2020*). Our samples presented bacterial constituents of human oral and soil microbiota at different proportions between the samples (*Figure 1—figure supplements 4–7*). Although no lethal bacterial pathogen was retrieved, some ancient dental pathogens (*Tannerella forsythia*) were reconstructed and described in more detail by *Bravo-Lopez et al., 2020* (*Figure 1—figure supplements 4–7*).

We verified the authenticity of the reads mapped to HBV (BWA) or B19V (BWA/blastn) in the enriched libraries (Materials and methods) by querying the reads against the non-redundant (nr) NCBI database using megaBLAST (*Altschul et al., 1990*). This step was performed to avoid including in the genome assembly reads that were mapped by BWA or blastn as HBV or B19V, but with a similar identity to a different taxon in the nr database (and absent in DS1; Materials and methods). Therefore, we only retained reads for which the top hit was to either B19V or HBV (*Supplementary file 1C*). To confirm the ancient origin of these viral reads, we evaluated the misincorporation damage patterns using the program mapDamage 2.0 (*Jónsson et al., 2013*), which revealed an accumulation of C to T mutations towards their 5' terminal site with an almost symmetrical G to A pattern on the 3' end (*Figure 2a*, *Figure 2—figure supplement 1*), as expected for aDNA (*Briggs et al., 2007*). Three ancient B19V genomes were reconstructed (*Figure 2b*, *Supplementary file 1C*) with sequence coverages between 92.37% and 99.1%, and average depths of 2.98–15.36× along their single-stranded DNA (ssDNA) coding region, which excludes the double-stranded DNA (dsDNA) hairpin regions at each end of the genome (*Luo and Qiu, 2015*). These dsDNA inverse terminal repeats (ITRs) displayed considerably higher depth values (<218×) compared to the coding region consistent with the better *postmortem* preservation of dsDNA compared to ssDNA (*Lindahl, 1993*; *Figure 2b*). In addition, we reconstructed one ancient HBV genome (*Figure 2c*, *Supplementary file 1C*) at 30.8× average depth and with a sequence coverage of 89.9%, including its ssDNA region at a reduced depth (<10×).

The reconstructed ancient HBV genome shows a 6 nucleotide (nt) insertion in the core gene, which is characteristic of the genotype A (*Kramvis, 2014*). Further phylogenetic analyses (Materials and methods) revealed that the Colonial HBV genome clustered with modern sequences corresponding to sub-genotype A4 (previously named A6) (*Pourkarim et al., 2014*; *Figure 3a*, *Figure 3—figure supplement 1*). The genotype A (HBV/GtA) has a broad diversity in Africa reflecting its long history in this continent (*Kostaki et al., 2018*; *Kramvis, 2014*), while the sub-genotype A4 has been recovered uniquely from African individuals in Belgium (*Pourkarim et al., 2010*) and has never been found in the Americas. Regarding the three Colonial B19V genomes from individuals HSJN240, COYC4, and HSJNC81 (C81A), these were phylogenetically closer to modern B19V sequences belonging to genotype 3 (*Figure 3b*, *Figure 3—figure supplements 2–3*). This B19V genotype is divided into two sub-genotypes: 3a, which is mostly found in Africa, and 3b, which is proposed to have spread outside Africa recently (*Hübschen et al., 2009*). The viral sequences from the individuals HSJN240 and COYC4 are similar to sub-genotype 3b genomes sampled from immigrants (Morocco, Egypt, and Turkey) in Germany (*Schneider et al., 2008*; *Figure 3b*, *Figure 3—figure supplements 2–3*) while the sequence of the individual HSJNC81 is more similar to a divergent sub-genotype 3a strain (*Figure 3b*, *Figure 3—figure supplements 2–3*) retrieved from a child with severe anemia born in France (*Nguyen et al., 1999*). These observations support the African origin of the reconstructed Colonial viral genomes.

In order to use our reconstructed viral genomes as molecular fossils to recalibrate each virus phylogeny and perform evolutionary inferences, we first needed to estimate if the phylogenetic relationships among B19V or HBV genomes had a temporal structure (i.e., sufficient genetic change

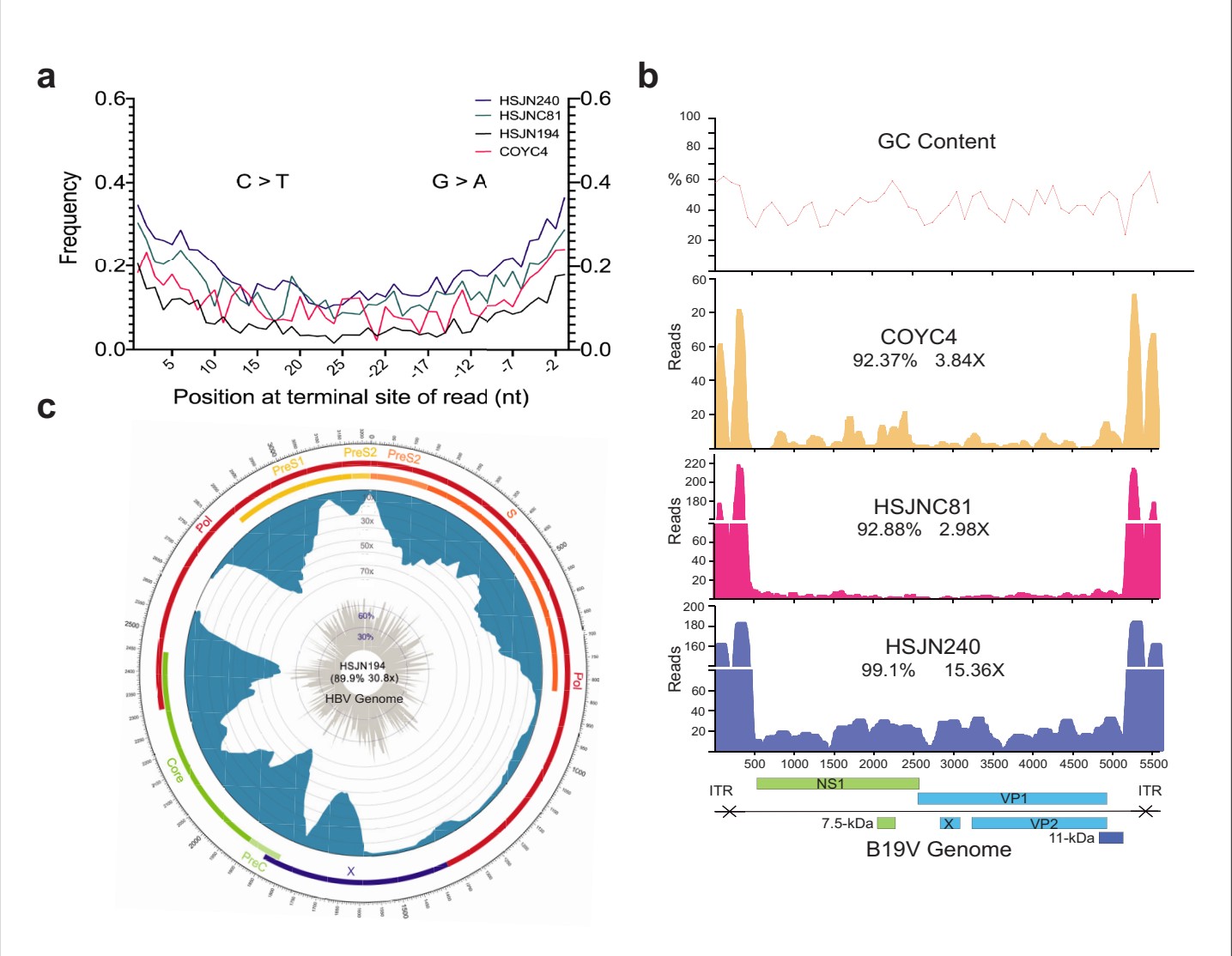

**Figure 2.** Ancient B19V and HBV ancient genomes. (**a**) Superimposed damage patterns of ancient HBV (HSJN194) and B19V (HSJNC81, HSJN240, COYC4). X-axis shows the position (nt) on the 5′ (left) and 3′ (right) end of the read, and Y-axis shows the damage frequency (raw individual damage patterns are shown in *Figure 2—figure supplement 1*). (**b**) B19V ssDNA linear genome. X-axis shows position (nt) based on the reference genome (AB550331), and Y-axis shows depth (as number of reads). GC content is shown as a percentage of each 100 bp windows. Coverage and average depth for the CDS are shown under each individual ID. Schematic of the B19V genome is shown at the bottom. Highly covered regions correspond to dsDNA ITRs shown as crossed arrows. (**c**) HBV circular genome. Outer numbers show position (nt) based on reference genome (GQ331046), outer bars show genes with names, blue bars represent coverage, and gray bars shows GC content each 10 bp windows. Coverage and average depth are shown in the center. Low covered region between S and X overlaps with ssDNA region.

The online version of this article includes the following figure supplement(s) for figure 2:

**Figure supplement 1.** Damage patterns of ancient viral genomes.

**Figure supplement 2.** Damage patterns of mitochondria.

between sampling times to reconstruct a statistical relationship between genetic divergence and time) (*Rambaut et al., 2016*). In the context of viruses, temporal structure is canonically tested with a root-to-tip distance and date randomization analyses (see *Firth et al., 2010*; *Rieux and Balloux, 2016*). Similarly to previous studies (*Krause-Kyora et al., 2018*; *Patterson Ross et al., 2018*), we found little or no temporal structure for this HBV phylogeny containing all genotypes ($R^2$ = 0.1351; correlation coefficient = 0.3676) (*Figure 3—figure supplement 5a-c*). The complex evolution of HBV may not be prone to an appropriate genetic dating since multiple inter-genotype recombination and

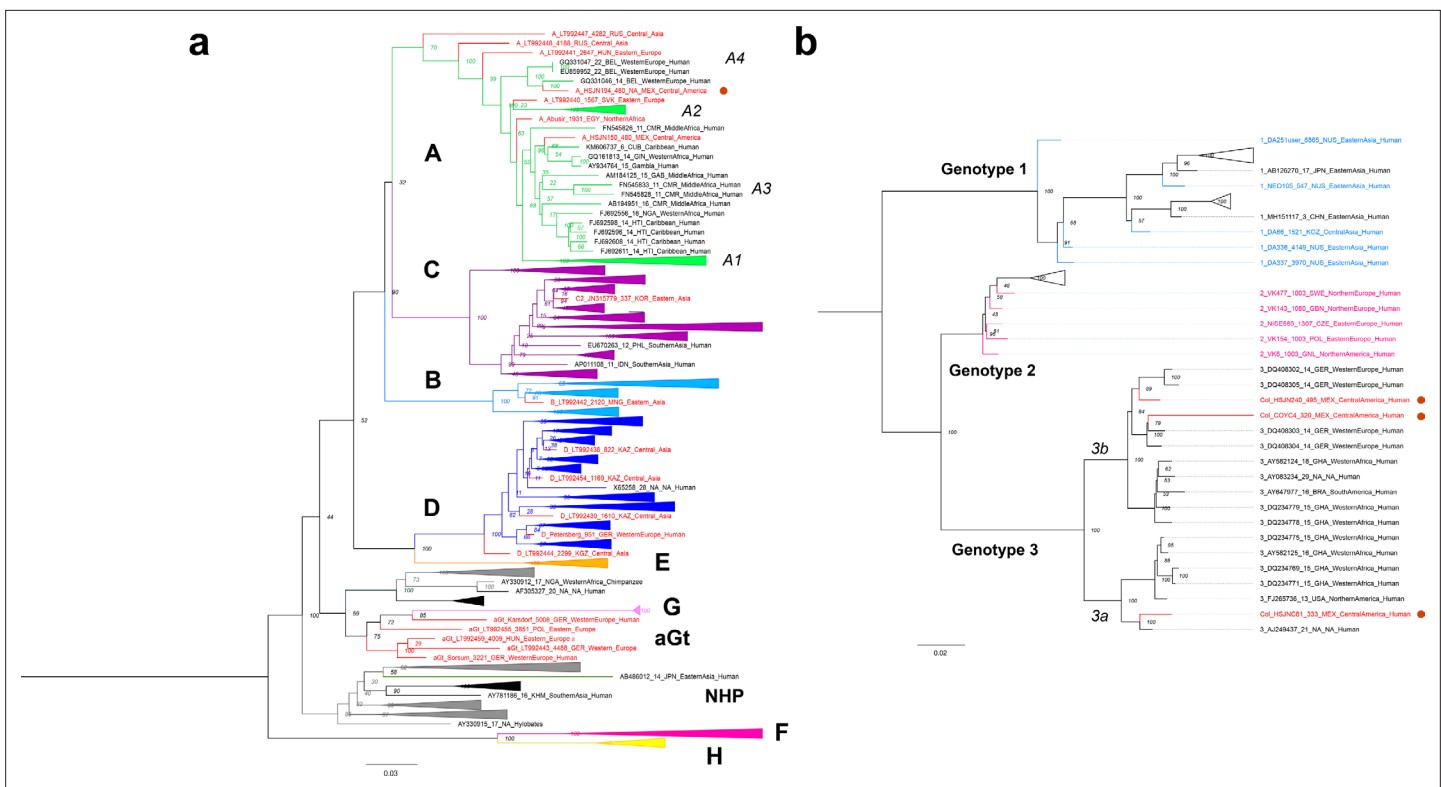

**Figure 3.** Viral Colonial genomes are similar to modern African genetic diversity. (**a**, **b**) Maximum likelihood trees performed on RAxML 8.2.10 (1000 bootstraps) with a midpoint root Genotypes are named in bold letters and sub-genotypes in italics. Bootstrap values are shown at the node center, and triangles represent collapsed sequences from other genotypes. Sequences are named as follows: genotype_ID_sampling.year_country.of.origin_area. of.origin_host. Sequences from this study are highlighted with a red circle on the right. (**a**) Based on the HBV whole genome, genotypes are named with letters and each is colored differently, while ancient sequences are shown in red. NHP: non-human primates. (**b**) Based on B19V CDS, genotypes are named with numbers, and only ancient genomes are colored.

The online version of this article includes the following figure supplement(s) for figure 3:

**Figure supplement 1.** Phylogenetic analysis of HBV.

**Figure supplement 2.** Neighbor-joining analysis of B19V.

**Figure supplement 3.** Dated coalescent phylogenetic analysis for B19V.

**Figure supplement 4.** Posterior probability densities of B19V dated coalescent phylogeny.

**Figure supplement 5.** Root-to-tip regression analysis of HBV temporal structure.

**Figure supplement 6.** Root-to-tip regression analysis of B19V temporal structure.

**Figure supplement 7.** Date randomization test B19V.

cross-species transmission (Human-Ape) events (***Krause-Kyora et al., 2018***) occurred throughout its evolution. Since the entire genotype A has been identified as a recombinant genotype before (***Mühlemann et al., 2018a***), we analyzed it independently and identified a stronger temporal signal within this genotype ($R^2$ = 0.722; correlation coefficient = 0.8498) (***Figure 3—figure supplement 5d-f***). In the case of B19V, we identified a temporal structure when including all three genotypes ($R^2$ = 0.3837; correlation coefficient = 0.6194) (***Figure 3—figure supplement 6a-c***), in agreement with previous studies (***Mühlemann et al., 2018b***). Furthermore, we corroborated this temporal structure was not an artifact by a set of tip-dated randomized analyses (***Rieux and Balloux, 2016***), where none of the 95% highest posterior density (HPD) intervals of the clock rate overlapped with the correctly dated dataset (***Figure 3—figure supplement 7***).

Given its strong temporal structure, we then performed a dated coalescent phylogenetic analysis for B19V (***Supplementary file 1D***). We inferred a median substitution rate for B19V of $1.03 \times 10^{-5}$ (95 % HPD: $8.66 \times 10^{-6}$–$1.21 \times 10^{-5}$) s/s/y under a strict clock and a constant population prior, and a substitution rate of $2.62 \times 10^{-5}$ (95 % HPD: $1.50 \times 10^{-5}$–$3.98 \times 10^{-5}$) s/s/y under a relaxed log normal

clock and a constant population prior. The divergence times from the most recent common ancestor of genotypes 1, 2, and 3 under a strict clock were 7.19 (95% HPD: 6.98–7.46), 2.11 (95% HPD: 1.83–2.51), and 3.64 (95% HPD: 3.04–4.33) ka, respectively. The inferred substitution rates and divergence times from the most recent common ancestor for genotypes 1 and 2 were similar to previous estimations (*Mühlemann et al., 2018b*) that included much older sequences, while the divergence of genotype 3 was subtly older since no other ancient genotype 3 had been reported previously.

Next, we used the shotgun data generated to determine the mitochondrial haplogroup of the hosts, as well as their autosomal genetic ancestry using the 1000 Genomes Project (*1000 Genomes Project Consortium, 2015*) as a reference panel (*Figure 4a*, *Supplementary file 2A*). The nuclear genetic ancestry analysis showed that all three HSJN individuals, from which the reconstructed viral genomes were isolated, fall within African genetic variation in a principal component analysis (PCA) plot (*Figure 4a*), while their mitochondrial aDNA belong to the L haplogroup, which has high frequency in African populations (*Supplementary file 2A*, *Figure 2—figure supplement 2*). Additionally, we performed $^{87}$Sr/$^{86}$Sr isotopic analysis on two of the HSJN individuals using tooth enamel as well as phalanx (HSJN240) or parietal bone (HSJNC81) to provide insights on the places of birth (adult enamel) and where the last years of life were spent (phalanx/parietal). The $^{87}$Sr/$^{86}$Sr ratios measured on the enamel of the individual HSJNC81 (0.71098) and HSJN240 (0.71109) are similar to average $^{87}$Sr/$^{86}$Sr ratios found in soils and rocks from West Africa (average of 0.71044, *Figure 4—figure supplement 1*, *Supplementary file 2*), as well as to $^{87}$Sr/$^{86}$Sr ratios described in first-generation Africans in the Americas (*Barquera et al., 2020*; *Bastos et al., 2016*; *Fricke et al., 2020*; *Price et al., 2012*; *Schroeder et al., 2009*). In contrast, the $^{87}$Sr/$^{86}$Sr ratios on the parietal and phalange bones from the HSJNC81 (0.70672) and HSJN240 (0.70755) show lower values similar to those observed in the Trans Mexican Volcanic Belt where the Mexico City Valley is located (0.70420–0.70550, *Figure 4—figure supplement 1*, *Supplementary file 2*). Moreover, radiocarbon dating of HSJN240 (1442–1608 CE, years calibrated for 1σ) and HSJN194 (1472–1625 CE, years calibrated for 1σ) (*Supplementary file 2A*, *Figure 4—figure supplement 2*) indicates that these individuals arrived during the first decades of the Colonial period, when the number of enslaved individuals arriving from Africa was particularly high (*Aguirre-Beltrán, 2005*).

Strikingly, Colonial individual COYC4, who was also infected with an African B19V strain, clusters with present-day Mexicans and Peruvians from the 1000 Genomes Project (*Figure 4a*). An ADMIXTURE (*Alexander and Lange, 2011*) analysis with these data confirmed a Native American genetic component (*Figure 4b*), as expected for an indigenous individual. The B19V ancient genome from the individual COYC4 is the first genotype 3 genome obtained from a non-African individual and suggests that following the introduction from Africa, the virus (B19V) spread and infected people of different ancestries during the Colonial period.

## Discussion

In this study, we reconstructed one HBV and three B19V ancient genomes from four different individuals using NGS, metagenomics, and in-solution targeted enrichment methods (*Figure 2b,c*, *Figure 1—figure supplement 1*). Several lines of evidence support the ancient nature of these viral sequences, in contrast to environmental contamination or a capture artifact. First, our negative control was not enriched for B19V or HBV hits in our capture sequencing (*Figure 1c*). For those samples that showed an enrichment in viral sequences after capture, the reads covered the reference genomes almost in their entirety and displayed deamination patterns at the terminal ends of the reads, as expected for aDNA (*Figure 2a*). Moreover, it is important to notice that B19V and HBV are blood-borne human pathogens that are not present in soil or the environment, and that DNA from these viruses had never been extracted before in the aDNA facilities used for this study.

The recovery of aDNA from B19V, which has a ssDNA genome (with dsDNA terminal repeats), in previous studies (*Mühlemann et al., 2018b*) as well as in our samples is noteworthy considering the NGS libraries were constructed using dsDNA as a template. Therefore, we would not expect to recover the ssDNA from B19V with this library building method. However, it is known that dsDNA intermediates form during the B19V replication cycle (*Ganaie and Qiu, 2018*), and that throughout the viral infection the replicating genomes are present in both the ssDNA and dsDNA forms. The sequences we retrieved must therefore correspond to the cell-free dsDNA replication intermediates. This is coherent with the peculiar coverage pattern on the B19V genome, where the dsDNA hairpins

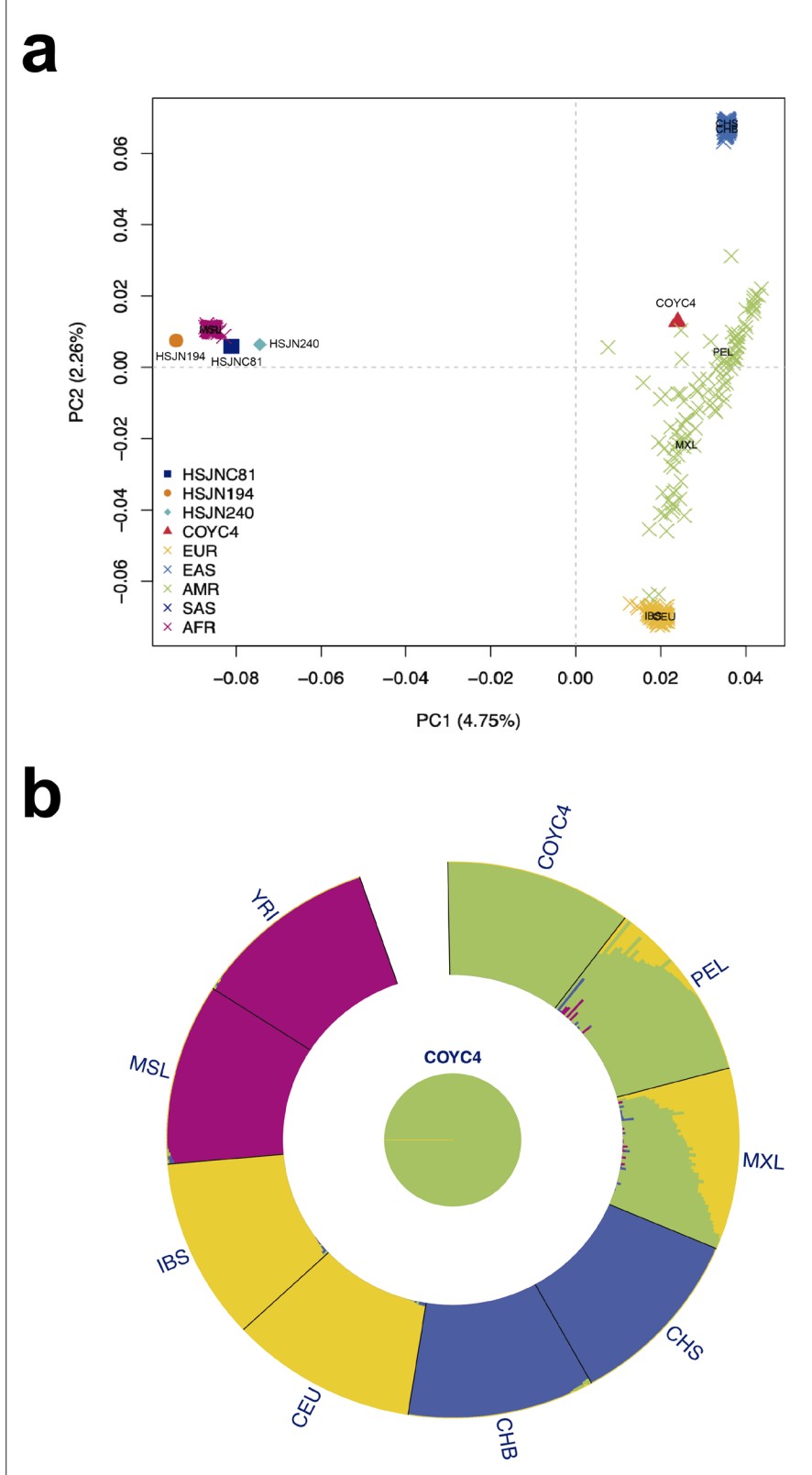

**Figure 4.** Human hosts are similar to modern African genetic diversity. (**a**) Principal component analysis (PCA) showing genetic affinities of ancient human hosts compared to the 1000 Genomes Project reference panel. Crosses (X) show individuals from the reference panel while other shapes show human hosts from which ancient HBV (HSJN194) and B19V (HSJNC81, HSJN240, COYC4) sequences were recovered. Clusters are colored in five

*Figure 4 continued on next page*

*Figure 4 continued*

super populations. EUR: Europeans (IBS, CEU); EAS: East Asian (CHB); AMR: Admixed populations from the Americas (MXL, PEL); SAS: South Asians (CHS); and AFR: Africans (YRI, MSL). Three-letter code is based on the 1000 Genomes Project nomenclature. (**b**) ADMIXTURE analysis with COYC4 intersected sites with 1000 Genomes MEGA array, run with k = 4 for 100 replicates. Each color shows a different component using the same colors as in the PCA. In the center, a pie chart shows the proportion of Native American (green).

The online version of this article includes the following figure supplement(s) for figure 4:

**Figure supplement 1.** [87] Sr/[86] Sr and Sr concentrations from HSJN individuals.

**Figure supplement 2.** Radiocarbon dating.

**Figure supplement 3.** Individuals from the HSJN positive for ancient viruses.

at its terminal sites and are highly covered, reflecting a better stability of these regions over time (*Figure 2b*). Similarly, the partially circular dsDNA genome from HBV was poorly covered at the ssDNA region (*Figure 2c*), which also goes through a dsDNA phase during replication, a similar coverage is reported in three previous ancient HBV genomes (*Krause-Kyora et al., 2018*). Although HBV and B19V are also capable of integrating into the human host genome (*Yuen et al., 2018*; *Janovitz et al., 2017*), the uneven read coverage observed for all reconstructed viruses (higher coverage in dsDNA regions) suggests that these sequences do not correspond to integration events. If the B19V or HBV reads we recovered derived from integrated sequences in the human genome, we would expect an even coverage along the reference viral genome, which is not the case. Further analyses would be needed to determine if the aDNA retrieved in this and other studies comes from systemic circulating virions or from systemic cell-free DNA intermediates (*Cheng et al., 2019*) produced after viral replication in the bone marrow or liver for B19V and HBV, respectively (*Broliden et al., 2006*; *Yuen et al., 2018*).

The ancient B19V genomes were assigned to genotype 3. This genotype is most prevalent in West Africa (Ghana: 100%, n = 11; Burkina Faso: 100%, n = 5) (*Candotti et al., 2004*; *Hübschen et al., 2009*; *Rinckel et al., 2009*) and a potential African origin has been suggested (*Candotti et al., 2004*). It has also been sporadically found outside of Africa (*Jain et al., 2015*) (*Candotti et al., 2004*; *Rinckel et al., 2009*) in Brazil (50%, n = 12) (*Freitas et al., 2008*; *Sanabani et al., 2006*), India (15.4%, n = 13) (*Jain et al., 2015*), France (11.4%, n = 79) (*Nguyen et al., 1999*; *Servant et al., 2002*), and the USA (0.85%, n = 117) (*Rinckel et al., 2009*) as well as in immigrants from Morocco, Egypt, and Turkey in Germany (6.7%, n = 59) (*Schneider et al., 2008*). Two other genotypes, 1 and 2, exist for this virus. Genotype 1 is the most common and is found worldwide, while the almost extinct genotype 2 is mainly found in elderly people from Northern Europe (*Pyöriä et al., 2017*). Ancient genomes from genotypes 1 and 2 have been recovered from Eurasian samples, including a genotype 2 B19V genome from a 10th-century Viking burial in Greenland (*Mühlemann et al., 2018b*). [87]Sr/[86]Sr isotopes on individuals from this burial revealed that they were immigrants from Iceland (*Mühlemann et al., 2018b*), suggesting an introduction of the genotype 2 to North America during Viking explorations of Greenland.

While serological evidence indicates that B19V currently circulates in Mexico, only the presence of genotype 1 has been formally described using molecular analyses (*Valencia Pacheco et al., 2017*). Taken together, our results are consistent with an introduction of the genotype 3 to New Spain as a consequence of the transatlantic slave trade imposed by European colonization. This hypothesis is supported by the [87]Sr/[86]Sr isotopic analysis, which suggests that the individuals from the HSJN with B19V (HSJN240, HSJNC81) were born in West Africa and spent their last years of life in New Spain (*Figure 4—figure supplement 1*). Furthermore, the radiocarbon analysis for individuals HSJN240 and HSJN194 (*Figure 4—figure supplement 2*) support this notion as they correspond to the Early Colonial period, during which the number of enslaved Africans arriving was higher compared to later periods (*Aguirre-Beltrán, 2005*). Remarkably, a B19V genome belonging to the genotype 3 was recovered from an individual (COYC4) with 100 % Indigenous ancestry (*Figure 4b*). COY4 was excavated in an independent archeological site 10 km south of the HSJN (*Figure 1a*), supporting the notion that viral transmissions between African individuals and Native Americans occurred during the Colonial period in Mexico City.

The HBV genotype A is highly diverse in Africa, reflecting its long evolutionary history, and likely originated somewhere between Africa, the Middle East, and Central Asia (*Kostaki et al., 2018*). The introduction of the genotype A from Africa to the Americas has been proposed based on phylogenetic analysis of modern strains from Brazil (*Freitas et al., 2008*; *Kostaki et al., 2018*) and Mexico (*Roman et al., 2010*), and more precisely of the sub-genotype A1 using sequences from Martinique (*Brichler et al., 2013*), Venezuela (*Quintero et al., 2002*), Haiti (*Andernach et al., 2009*), and Colombia (*Alvarado-Mora et al., 2012*). Recently, a similar introduction pattern was proposed for the quasi genotype A3 based on an ancient HBV genome recovered from an ancient African individual sampled in Mexico (*Barquera et al., 2020*). The origin of the sub-genotype A4 is controversial since the apparent African origin is based on modern sequences recovered from African immigrants living in Europe (*Pourkarim et al., 2010*). The Colonial ancient HBV genome reconstructed in our work represents the first ancient A4 linked to the transatlantic slave trade (*Figure 3a*, *Figure 3—figure supplement 1*), and the only report of this sub-genotype in the Americas, further supporting its African origin. The introduction of pathogens from Africa to the Americas has been proposed for other human-infecting viruses such as smallpox (*Mandujano-Sánchez et al., 1982*; *Somolinos d'Árdois, 1982*), based on historical records; or yellow fever virus (*Bryant et al., 2007*), HTLMV-1 (*Gadelha et al., 2014*), hepatitis C virus (genotype 2) (*Markov et al., 2009*), and human herpes simplex virus (*Forni et al., 2020*) based on phylogenetic analysis of modern strains from Afro-descendant or admixed human populations.

Although we cannot assert where exactly the African-born individuals in this study contracted B19V or HBV (Africa, America, or the Middle Passage) nor if the cause of their deaths can be attributed to such infections, the identification of ancient B19V and HBV in contexts associated with Colonial epidemics in Mexico City is still relevant in light of their paleopathological marks and the clinical information available for the closest sequences in the phylogenetic analyses. Notably, individual HSJNC81 displayed cribra orbitalia in the eye sockets and porotic hyperostosis on the cranial vault (*Figure 4—figure supplement 3*). The reconstructed ancient B19V genome from this individual is closest to the V9 strain, which was isolated from an infant with severe anemia and G6PD deficiency (*Nguyen et al., 1999*; GenBank: AJ249437; *Figure 3b*). The HSJN skeletal collection has a notably higher rate of cribra orbitalia and porotic hyperostosis compared to other Colonial archeological sites, marks that were proposed to be caused by an unknown infectious disease (*Castillo-Chavez, 2000*). These skeletal indicators are caused by irregular hematopoiesis in the bone marrow and are typically associated with genetic anemias such as thalassemia and sickle cell anemia (*Angel, 1966*), as well as to nutritional stress or parasitic infections (*Walker et al., 2009*). It is acknowledged that B19V infection can cause severe or even fatal anemia due to the low level of hemoglobin in individuals with other blood disorders, such as thalassemia, sickle cell anemia, malaria, or iron deficiency (*Broliden et al., 2006*; *Heegaard and Brown, 2002*). Therefore, since B19V infects precursors of the erythroid lineage (*Broliden et al., 2006*), it is possible that the morphological changes found in HSJNC81 might be the result of a severe anemia caused or enhanced by a B19V infection. With our data we cannot discard the simultaneous presence of a genetic disease since the loci for thalassemia, sickle cell anemia, and G6PD deficiency were not covered with our human-mapped NGS data. Nevertheless, the identification of ancient B19V in a Colonial context is noteworthy considering several recent reports reveal that measles-like cases were actually attributable to B19V (*De Los Ángeles Ribas et al., 2019*; *Rezaei et al., 2016*) or rubella (*Anderson et al., 1985*; *Davidkin et al., 1998*; *De Los Ángeles Ribas et al., 2019*; *Rezaei et al., 2016*), which produce a similar kind of rash and fever. Therefore, it is possible that B19V might have been responsible for some of the numerous cases attributed to measles that were described in early 16th-century Mexico (*Acuña-Soto et al., 2004*; *Mandujano-Sánchez et al., 1982*; *Wesp, 2017*), in particular historical records that document the treatment of an outbreak of measles at the HSJN in 1531 CE (*Meza, 2013*). Our study, however, does not reject the notorious role that measles played during the Colonial outbreaks (as it is strongly supported by historical records), but provides evidence of the presence of B19V during the Colonial period in Mexico City to facilitate discussions about the paradigmatic etiology of the supposed measles epidemics reported in historical records (*Malvido and Viesca, 1982*; *Mandujano-Sánchez et al., 1982*; *Somolinos d'Árdois, 1982*). This hypothesis requires additional comprehensive studies aimed at characterizing the presence of measles and rubella viruses from ancient remains, a task that currently poses difficult technical challenges given that RNA is known to degrade rapidly. In fact,

most ancient viral RNA genomes have been recovered only from formalin-fixed tissue (*Düx et al., 2020*; *Xiao et al., 2013*).

Furthermore, historical records of the autopsies of the victims of the 1576 CE *Cocoliztli* epidemic treated at the HSJN describe the presence of enlarged hard liver and jaundice (*Acuña-Soto et al., 2002*; *Acuña-Soto et al., 2004*; *Malvido and Viesca, 1982*; *Marr and Kiracofe, 2000*; *Somolinos d'Árdois, 1982*) as well as a black spleen and lungs and heart with yellow liquid and black blood (*Acuña-Soto et al., 2000*; *Malvido and Viesca, 1982*; *Somolinos d'Árdois, 1982*). This is noteworthy given that both HBV and B19V viruses proliferate in the liver and are associated with hepatitis and jaundice (*Broliden et al., 2006*; *Yuen et al., 2018*). The radiocarbon dating of individuals HSJN194 (HBV) and HSJN240 (B19V) suggests that these individuals died between 1472–1625 CE and 1442–1608 CE (years calibrated for 1σ), respectively (*Figure 4—figure supplement 2*), which overlaps with the period of time when the hepatitis symptoms were reported in the autopsies after the 1576 *Cocoliztli* epidemic at the HSJN (*Acuña-Soto et al., 2004*; *Marr and Kiracofe, 2000*; *Somolinos d'Árdois, 1982*). However, additional analyses are needed before being able to establish a link between these viruses and the wide array of symptoms described for *Cocoliztli*. Currently, technological limitations prevent the direct identification of ancient RNA viruses in bone or dental remains. However, future studies, with larger sample sizes from different contexts associated with the outbreak, should explore a wider range of pathogens previously suggested as potential causative agents, like arthropod-borne pathogens (malaria, yellow fever virus, and dengue virus) (*Marr and Kiracofe, 2000*) or hemorrhagic fever RNA viruses (*Acuña-Soto et al., 2004*).

Furthermore, it is important to acknowledge that both viruses have also been previously identified in aDNA datasets not necessarily associated with disease or epidemic contexts (*Kahila Bar-Gal et al., 2012*; *Krause-Kyora et al., 2018*; *Mühlemann et al., 2018a*; *Patterson Ross et al., 2018*). Additionally, our data is not sufficient to elucidate the age when the individuals acquired the viruses or if it is related to their cause of death.

In the case of HSJN194, we cannot establish if he acquired HBV vertically or horizontally, nor if this individual presented an acute or chronic infection. Finally, although our data does not allow us to associate these viruses to a specific epidemic outbreak, the identification of HBV and B19V in Post-Contact remains opens up new opportunities for investigating their presence in similar contexts and expand our knowledge on their evolution and potential link to disease in Colonial Mexico. This type of research is particularly relevant when considering previous hypotheses favoring the synergistic action of different types of pathogens in these devastating Colonial epidemics (*Somolinos d'Árdois, 1982*).

It is important to emphasize that our findings should be interpreted with careful consideration of the historical and social context of the transatlantic slave trade. This cruel episode in history involved the forced displacement of millions of individuals to the Americas (ca. 250,000 to New Spain; *Aguirre-Beltrán, 2005*) under inhumane, unsanitary, and overcrowded conditions that, with no doubt, favored the spread of infectious diseases (*Mandujano-Sánchez et al., 1982*). Therefore, the introduction of these and other pathogens from Africa to the Americas should be attributed to the brutal and harsh conditions of the Middle passage that enslaved Africans were subjected to by traders and colonizers, and not to the African peoples themselves. Moreover, the adverse life conditions for enslaved Africans and Native Americans, especially during the first decades after colonization, surely favored the spread of diseases and emergence of epidemics (*Mandujano-Sánchez et al., 1982*). Integrative and multidisciplinary approaches are thus needed to understand this phenomenon in full.

In summary, our study provides direct aDNA evidence of HBV and B19V introduced to the Americas from Africa during the transatlantic slave trade. The isolation and characterization of these ancient HBV and B19V genomes represent an important contribution to the ancient viral genomes reported in the Americas (*Barquera et al., 2020*; *Duggan et al., 2020*; *Schrick et al., 2017*). Our results expand our knowledge on the viral agents that were in circulation during Colonial epidemics like *Cocoliztli*, some of which resulted in the catastrophic collapse of the immunologically naïve Indigenous population. Although we cannot assign a direct causality link between HBV and B19V and *Cocoliztli*, our findings confirm that these potentially harmful viruses were indeed circulating in individuals found in archeological contexts associated with this epidemic outbreak. Further analyses from different sites and samples will help understand the possible role of these and other pathogens in Colonial epidemics, as well as the full spectrum of pathogens that were introduced to the Americas during European colonization.

## Materials and methods

### Sample selection and DNA extraction

Dental samples (premolars and molars) were obtained from 21 individuals from the skeletal collection associated with the HSJN and were selected based on morphological features indicating a possible African origin (*Hernández-Lopez and Negrete, 2012*; *Karam-Tapia, 2012*; *Meza, 2013*; *Ruíz-Albarrán, 2012*). Five additional samples were taken from 'La Concepción' chapel, based on their conservation state. Permits 401.1 S.3-2018/1373 and 401.1 S.3-2020/1310 to carry out this sampling and aDNA analyses were obtained from the Archeology Council of the National Institute of Anthropology and History (INAH) for the Hospital San Jose de los Naturales and 'La Concepción' chapel, respectively. Two of the individuals from whom the ancient viral genomes were retrieved (HSJN194 and HSJN240) are mostly complete articulated skeletons and one individual (HSJNC81) is an isolated cranium recovered during the early excavation stage and does not have any associated postcranial elements. The archeologists suggest that all of the individuals were deposited during an infectious disease epidemic in a mass burial context (*Figure 1b*; *Cabrera-Torres and García-Martínez, 1998*).

### DNA extraction and NGS library construction

Bone samples were transported to a dedicated ancient DNA clean-room laboratory at the International Laboratory for Human Genome Research (LIIGH-UNAM, Querétaro, Mexico), where DNA extraction and NGS library construction was performed under the guidelines on contamination control for aDNA studies (*Warinner et al., 2017*). Teeth were carefully cleaned with NaClO (70%) and ethanol (70%) superficially and later exposed to UV light for 1.5 min. The tooth root was sectioned from the crown and fragmented by mechanical pressure. Previously reported aDNA extraction protocols were used on approximately 200 mg of tooth root powder obtained from the HSJN and COY samples (*Dabney et al., 2013*; *Rohland and Hofreiter, 2007*). A blank extraction control per batch was used to identify the presence of environmental and cross-sample contamination. dsDNA indexed (6 bp) sequencing libraries were constructed using 30 µl of the DNA extract, as previously reported (*Meyer and Kircher, 2010*).

### Next-generation sequencing

Pooled libraries were sequenced on an Illumina NextSeq550 at the 'Laboratorio Nacional de Genómica para la Biodiversidad' (LANGEBIO, Irapuato, Mexico), with a mid-output 2 × 75 format (paired-end). The reads obtained (R1 and R2) were merged (>11 bp overlap) and trimmed with AdapterRemoval 1.5.4 (*Schubert et al., 2016*). Overlapping reads (>30 bp in length, quality filter >30) were kept and mapped to the human genome (hg19) using BWA 0.7.13 (aln Algorithm) (*Li and Durbin, 2009*). Mapped reads were used for further human analysis (genetic ancestry and mitochondrial haplogroup determination), whereas unmapped reads were used for metagenomic analysis and viral genome reconstruction.

### Metagenomic analyses

The Viral RefSeq database was downloaded from the NCBI ftp server on February 2018; this included 7530 viral genomes, including human pathogens. MALT 0.4.0 (*Herbig et al., 2018*) software was used to taxonomically classify the reads using the viral genomes database as a reference. The viral database was formatted automatically with malt-build once, and non-human (unmapped) reads were aligned with malt-run using the blastn and SemiGlobal mode with an 85 minimal percent identity (--minPercentIdentity) and e-values of 0.001 (--e). The RMA files were used for the normalization of the viral abundances based on the library with the smallest number of reads (default, (*count of class/total count of sample)* count of smallest sample*) and compared to all the samples from the same archeological site with MEGAN 6.8.0 (*Huson et al., 2016*).

Independently, unmapped reads (non-human) were taxonomically classified with Kraken2 (*Wood et al., 2019*) using a reference database composed of NCBI RefSeq bacterial, archaea, and viral genomes (downloaded on November 3, 2017). The Kraken2 output was transformed to a BIOM-format table using Kraken-biom (https://github.com/smdabdoub/kraken-biom; *Dabdoub et al., 2018*) and then visualized with Pavian (*Breitwieser and Salzberg, 2020*). Detailed description of the results can be found in *Bravo-Lopez et al., 2020*.

## In-solution enrichment assay design

Twenty-nine viruses were included in the design of biotinylated probes (*Supplementary file 1A*), including viral genomes previously recovered from archeological remains like B19V, B19V-V9, and HBV (consensus genomes), selected VARV genes, as well as clinically important viral families that are able to integrate into the human genome, have dsDNA genomes, or dsDNA intermediates.

The HBV majority consensus genome (>50% conservation per site) was constructed using an alignment of modern references (A–H genotype) and a well-covered (>5×  coverage) ancient genotype (*Mühlemann et al., 2018a*; LT992459).

Thirty VARV genes were chosen for a consensus sequence construction based on three categories; replication (J6R, A24R, A29L, E4L, A50R, A5R, D7R, H4L, E9L), structural (A27L, A25, D8L, H3L, L1R, A33R, B5R, A16L), and immune host regulation (B18R, A46R, B15R, K7R, N1L, M2L, E3L, H1L, B8R, D9R, D10, K3L), and were obtained from all the available VARV genomes including three ancient genomes (NCBI GenBank 2019 *Duggan et al., 2016*; *Pajer et al., 2017*). The selected genes were aligned in AliView (MUSCLE algorithm *Edgar, 2004*; *Larsson, 2014*) to generate a majority consensus for every gene. The generated consensus sequences targeted <20% of the VARV whole genome.

For the *Herpesviridae* family, a total of 19 genes were selected, 6 from herpes simplex virus 1 (UL30, UL31, UL19, UL27, US6, UL10), 6 from human cytomegalovirus (UL54, UL53, UL86, UL115, UL75, UL83), and 7 from Epstein–Bar virus (ORF9, ORF69, ORF25, ORF47, ORF8, vIRF2, K5). GenBank IDs are shown in *Supplementary file 1A*.

Selected genes from VARV and *Herpesviridae* were defined as 40 bp or 60 bp upstream the start codon, and downstream the stop codon, respectively, in order to ensure a uniform coverage of the entire coding region in case of a positive sample.

The resulting design comprised 19,147 ssRNA 80 nt probes targeting, with a 20 nt interspaced distance, the whole or partial informative regions of eight viral families (*Poxviridae, Hepadnaviridae, Parvoviridae, Herpesviridae, Retroviridae, Papillomaviridae, Polyomaviridae, Circoviridae*). To avoid a simultaneous false-positive DNA enrichment, low-complexity regions and human-like (hg38) sequences were removed (in silico). The customized kit was produced by Arbor Biosciences (Ann Arbor, MI, USA).

## Capture-enrichment assay

Capture-enrichment was performed on 30–90 ng (depending the availability) of the indexed libraries to pull-down viral aDNA using 60 °C during 48 hr for hybridization, based on the manufacturer's protocol (version 4). Libraries were amplified with 18–20 cycles (Phusion U Hot Start DNA Polymerase by Thermo Fischer Scientific) using primers for the adaptors of each post-capture library. PCR products were purified with SPRISelect Magnetic Beads (Beckman Coulter) and quantified with a Bioanalyzer 2100 (Agilent). Amplified libraries were then pooled in different concentrations and deep sequenced on an Illumina NextSeq550 (2 × 75 middle output) yielding >1 × 10$^6$ non-human reads (*Supplementary file 1C*). In order to saturate the target viral genome, one or two non-consecutive rounds of capture were performed for HBV and B19V, respectively. Reads generated from each enriched library were analyzed exactly as the shotgun (not-enriched) libraries. Normalized abundances between shotgun and captured libraries were compared in MEGAN 6.8.0 (*Huson et al., 2016*) to evaluate the efficiency and specificity of the enrichment assay.

## Viral datasets
### HBV-Dataset-1 (HBV_DS1)

This comprises 38 HBV genomes from modern A–J human genotypes, 2 well-covered ancient HBV genomes, and a wholly monkey genome. Genotype A: HE974381, HE974383, AY934764, GQ331046. Genotype B: B602818, AB033555, AB073835, AB287316, AB241117. Genotype C: AB111946, X75656, AB048704, AF241411, AP011102, AP011106, AP011108, AB644287. Genotype D: FJ899792, GQ477453, JN688710, HE974373, FJ904430, AB033559. Genotype E: HE974384. Genotype F: AY090458, AB116654, AY090455, DQ899144, HE974369, AB116549, AF223962, AB166850. Genotype G: AP007264. Genotype H: AB516395. Genotype J: AB486012. Ancient: LT992443, LT992459. Outgroup (Woolly Monkey): AF046996.

### HBV-Dataset-2 (HBV_DS2)

This comprises 593 whole genomes downloaded from the NCBI database in August 2020 that included the union of curated datasets used in four previous studies (*Drexler et al., 2013*; *Krause-Kyora et al., 2018*; *Mühlemann et al., 2018a*; *Paraskevis et al., 2015*), from which only non-duplicated HBV genomes were considered. This dataset contains genomes from A–J genotypes as well as non-human primate HBV genomes (gibbon, gorilla, and chimpanzee). Additionally, 19 ancient HBV genomes (*Barquera et al., 2020*; *Kahila Bar-Gal et al., 2012*; *Krause-Kyora et al., 2018*; *Mühlemann et al., 2018a*; *Neukamm et al., 2020*; *Patterson Ross et al., 2018*) and 1 ancient HBV genome from this study (HSJN194) were included.

### HBV_DS2.1

56 genomes assigned to genotype A, based on our ML analysis plus the ancient genome from the present study (HSJN194).

### B19V-Dataset-1 (B19V_DS1)

This comprises 13 B19V sequences of genotypes 1–3: KT268312, AY504945, FJ591158, EF216869, AY064476, DQ333427, AB550331, AY582124, DQ408305, FJ265736, AJ249437, NC_004295, NC_0008831, plus one outgroup (Bovine Parvovirus): NC_001540.

### B19V-Dataset-2 (B19V_DS2)

All B19V genomes retrieved from the NCBI database were downloaded (August 2020) using the next search command "*human parvovirus b19[organism] not rna[title] not clone[title] not clonal[title] not patent[title] not recombinant[title] not recombination[title] and 3000:6000[sequence length]*," which considers only whole genomes (3–6 kb), resulting in a total of 109 B19V genomes from genotypes 1–3. This dataset included the 10 best-covered ancient genomes from genotypes 1 and 2 (*Mühlemann et al., 2018b*) as well as 3 ancient B19V from this study. Since many of the reported genomes in our dataset are not complete, only the whole coding region (CDS) was used for phylogenetic analyses.

## Genome reconstruction and authenticity

### HBV

Non-human reads were simultaneously mapped to HBV_DS1 with BWA (aln algorithm) with seedling disabled (-l 1050) (*Schubert et al., 2012*). The reference sequence with the most hits was used to map uniquely to this reference and generate a BAM alignment without duplicates (ref: GQ331046), from which damage patterns were determined and damaged sites rescaled using mapDamage 2.0 (*Jónsson et al., 2013*). The rescaled alignment was used to produce a consensus genome. All the HBV mapped reads were analyzed through megaBLAST (*Altschul et al., 1990*) using the whole NCBI nr database to verify if they were assigned uniquely to HBV (carried out with Krona 2.7; *Ondov et al., 2011*).

### B19V

The reconstruction of the B19V ancient genome was done as previously reported from archeological skeletal remains (*Mühlemann et al., 2018b*), but with increased stringency of some parameters, which are described here. Non-human reads were mapped against B19V_DS1 with BWA (aln algorithm) with seedling disabled (*Schubert et al., 2012*). If more than 50 % of the genome was covered, the sample was considered positive to B19V. Reads from the B19V-positive libraries were aligned with blastn (-evalue 0.001) to B19V_DS1 to recover all the parvovirus-like reads. To avoid local alignments, only hits covering >85% of the read were kept and joined to the B19V mapped reads (from BWA). Duplicates were removed. The resulting reads were analyzed with megaBLAST (*Altschul et al., 1990*) using the whole NCBI nr database to verify the top hit was to B19V (carried out with Krona 2.7; *Ondov et al., 2011*). This pipeline was applied for two independent enrichments assays per sample and the filtered reads from the two capture rounds were joined. The merged datasets per sample were mapped using as a reference file the three known B19V genotypes with GeneiousPrime 2019.0.4 (*Kearse et al., 2012*) using median/fast sensibility and iterate up to five times. The genotype with the longest covered sequence was selected as the reference for further analysis (ref: AB550331).

Deamination patterns for HBV and B19V were determined with mapDamage 2.0 (*Jónsson et al., 2013*) and damaged sites were rescaled in the same program to produce a consensus whole genome using SAMtools 1.9 (*Li et al., 2009*).

## Phylogenetic analyses

HBV_DS2 and B19V_DS2 were aligned independently in Aliview (*Larsson, 2014*; MUSCLE algorithm; *Edgar, 2004*) and curated manually to have the same lengths. The alignments were evaluated in jModelTest 2.1.10 (*Darriba et al., 2012*) using a corrected Akaike information criterion (AICc) and Bayesian information criterion (BICc) tests that supported with 100 % confidence the evolutionary models used in our maximum likelihood analysis in RAxML (*Stamatakis, 2014*).

To test the temporal structure of our ML trees, a root-tip-dated analysis was performed on Tempest 1.5.3 (*Rambaut et al., 2016*) for both DS2 (B19V, HBV) in the presence or absence of ancient sequences and without the sequences presented in this study (*Figure 3—figure supplements 5–6*). In the case of HBV, an additional analysis was performed only on the genotype A to find a higher temporal structure in the presence or absence of ancient sequences and without the HSJN194 HBV genome presented in this study (*Figure 3—figure supplement 5*). For the B19V_DS2, the temporal structure suggested by root-tip distance analysis was corroborated using a date randomization test (DRT) with TipDatingBeast 1.0.5 (*Rieux and Khatchikian, 2017*) and BEAST 2.5.1 (*Drummond et al., 2012*; *Figure 3—figure supplement 6*).

Since the DRT and the root-tip-dated analysis suggested a temporal structure for the B19V_DS2, a coalescent dated tree was generated in BEAST 2.5.1 (*Drummond et al., 2012*) for B19V using a relaxed and strict clock; both with different priors (coalescent constant, exponential, and Bayesian skyline population priors), with an a priori substitution rate interval of $1 \times 10^{-3}$–$1 \times 10^{-7}$ s/s/y (*Mühlemann et al., 2018b*). For the Colonial genomes used in this study, a uniform sampling was indicated using the radiocarbon dates for HSJN240 (495 ± 166 ybp). When radiocarbon dating was not possible, an archeological date interval was set for HSJNC81 (332.5 ± 269 ybp) and COYC4 (320 ± 400 ybp), based on the archeological estimates of both sites. The strict molecular clock analyses were performed with a 50 million MCMC sampled each 5000 generations, while the relaxed molecular clock with exponential population was run with a 250 million MCMC sampled each 5000 generations, and the relaxed molecular clock with coalescent constant and Bayesian Skyline population priors were run with 250 million MCMC and with 350 million MCMC sampled each 5000 generations. Both files were mixed with a 25 % burn-in LogCombiner (*Drummond et al., 2012*). All the Bayesian analyses were mixed and reached convergence (>200 ESS) as estimated in Tracer 1.7 (*Rambaut et al., 2018*; *Supplementary file 1D*). The first 25 % of the generated trees were discarded (burn in) and a Maximum Clade Credibility Tree with median ages was created with TreeAnnotator (*Drummond et al., 2012*; *Figure 3—figure supplements 3–4*).

## Radiocarbon dating

Radiocarbon analysis was conducted at the Physics Institute of the National Autonomous University of Mexico (UNAM) for the individuals in this study with complete skeletons (HSJN194 and HSJN240). From these individuals, phalanx bones (left hand) were cleaned, dried, and powdered to be digested in a HCl 0.5 M solution followed by a NaOH 0.01 M and HCl 0.2 M treatment. Collagen was then filtered (>30 kDa) and graphitization was performed on an AGEIII (Ion Plus). $^{14}$C, $^{13}$C, and $^{12}$C isotopes were analyzed from graphite in a Tandetron (High Voltage Engineering Europa B.V.) mass spectrometer with a 1 V energy accelerator. Radiocarbon dates were estimated based on InCal13 (*Reimer et al., 2013*) calibration curve and corrected with OxCal v4.2.4. (*Bronk Ramsey, 2013*).

## Sr isotope analysis

Tooth enamel was carefully extracted with the aid of dental tools. The material underwent several cleaning procedures before crushing to a 50 μm grain size with an agate mortar. Chemicals used for this purpose included 30 % $H_2O_2$ and 1–1.5 N $HNO_3$. In between, rinses were performed with deionized water (Milli-Q). Ultrasonic bath (USB) was used to accelerate these processes. After obtaining the desired grain size, samples were treated with 30 % $H_2O_2$, 1 N $NH_4Cl$, and alternated with water washes. To get rid of any secondary contaminant or any postmortem external agent that could alter the Sr isotopic values, tooth samples were treated with a three-step leaching technique: the first

leachate is obtained with 0.1 N acetic acid for 30 min (USB). The solution is decanted and dried under infrared light (Lix 1). The residue was leached for 15 min in 1 N acetic acid (USB) and subsequently stored overnight for 12 hr in the same acid. The solution was decanted and dried to obtain the second leachate (Lix 2). The residue (Res) is dissolved in 8 N $HNO_3$ as well as Enamel Lix 1 and Enamel Lix 2 in closed Teflon beakers on a hot plate at 90 °C. A total of three aliquots from each molar were obtained from this leaching process. After sample digestion, Sr from teeth and bone samples was extracted with Sr-Spec (EICHROM) ion exchange column chemistry. Detailed analytical procedures are described in *Solís-Pichardo et al., 2017*. Sr isotope analysis was carried out with a Triton Plus (Thermo Scientific) thermal ionization mass spectrometer with 9 Faraday collectors at the 'Laboratorio Universitario de Geoquímica Isotópica' (LUGIS, UNAM). Sr was measured as metallic ions with 60 isotopic ratios that were normalized for mass fractionation to $^{86}Sr/^{88}Sr = 0.1194$. The mean value for the NBS 987 Sr standard was $^{87}Sr/^{86}Sr = 0.710254 \pm 0.000012$ ($\pm 1$ $sd_{abs}$, n = 86) and the analytical blank yielded 0.23 ng Sr. $^{87}Sr/^{86}Sr$ ratios were performed on the tooth enamel (crowns) of individuals HSJNC81 and HSJN240. Similar analyses were done for HSJNC81 and HSJN240 using the parietal and phalanx bone, respectively. In the case of the two individuals analyzed in this study, bone $^{87}Sr/^{86}Sr$ values 0.70672 (HSJNC81) and 0.70755 (HSJN240) (Table S6) are comparable to those obtained from soil samples from the eastern TMVB rim in Veracruz with a mean $^{87}Sr/^{86}Sr$ of 0.70703 (n = 6) (*Solís-Pichardo et al., 2017*). For West African igneous and metamorphic rocks, a mean value $^{87}Sr/^{86}Sr$ of 0.71044 was obtained (n = 20, *Figure 4—figure supplement 1*). Data are compiled in *Supplementary file 2* with their corresponding references.

## Principal component analysis

Human-mapped reads (BWA aln) obtained from the pre-capture sequence data of viral-positive samples were used to infer the genetic ancestry of the hosts using PCA. The genomic alignments (to hg19) of the four ancient individuals (HSJNC81, HSJN240, HSJN194, and COYC4) was intersected with the genotype data of 400 present-day individuals from eight populations (50 individuals per population) in the 1000 Genomes Project (*1000 Genomes Project Consortium, 2015*; IBS: Iberian from Spain; CEU: Utah Residents with Northern and Western European Ancestry; CHB: Han Chinese in Beijing; CHS: Southern Han Chinese, YRI: Yoruba in Ibadan; MSL: Mende in Sierra Leone, MXL: Mexican Ancestry from Los Angeles; and PEL: Peruvians from Lima; *Supplementary file 2A*). Pseudo haploid genotypes were called by randomly selecting one allele at each intersected site, both in the reference panel and in the genomic alignments, and filtering by a base quality >30 in the latter. The merged dataset was processed using PLINK (*Purcell et al., 2007*) with the following parameters: a linkage disequilibrium filter (--indep-pairwise 200 25 0.2), genotype missingness filter of 5 % (--geno 0.05), and minor allele frequency of 5 % (--maf 0.05). This resulted in 904,258 SNVs passing the filters. PCA was then performed on with the program smartpca (EIGENSOFT package) (*Patterson et al., 2006*; *Price et al., 2006*) using the option *lsqproject* to project the ancient individuals into the PC space defined by the modern individuals.

## Ancestry composition of individual COYC4

A total of 58,670 SNPs intersected between the 1000 Genomes Project reference panel and the COYC4 ancient genome (see previous section for details). The program ADMIXTURE (*Alexander and Lange, 2011*) was run on these intersected data with K values between 2 and 5, and 100 replicates for each K using a different random seed number. For each K, the ADMIXTURE run with the best likelihood was chosen to be plotted using AncestryPainter (*Feng et al., 2018*).

## Mitochondrial haplogroup and sex determination

NGS reads were mapped to the human mitochondrial genome reference (rCRS) with BWA (aln algorithm, -l default), the alignment file was then used to generate a consensus mitochondrial genome with the program Schmutzi (*Renaud et al., 2015*) The assignment of the mitochondrial haplogroup was carried out with Haplogrep (*Kloss-Brandstätter et al., 2011*; *Weissensteiner et al., 2016*) using the consensus sequence as the input. Assignment of biological sex was inferred based on the number of reads mapped to the Y-chromosome (Ry) relative to those mapping to the Y and X-chromosome (*Skoglund et al., 2013*). Ry <0.016 and Ry >0.075 were considered XX or XY genotype, respectively. The resulting XY sex was coherent with the one inferred morphologically (*Supplementary file 2A*).

## Acknowledgements

This work was funded by the Wellcome Trust Sanger grant number 208934/Z/17/Z, project IA201219 PAPIIT-DGAPA-UNAM, The Human Frontier Science Program number RGY0075/2019, and the Vaccine and Infectious Disease Division at Fred Hutchinson Cancer Research Center. DB-M is an Open Philanthropy Fellow of the Life Sciences Research Foundation (LSRF). We thank INAH's Consejo de Arqueología for approving the sampling and aDNA analysis (permits 401.1 S.3-2018/1373 and 401.1 S.3-2020/1310 for Hospital San Jose de los Naturales and the Temple of Immaculate Conception (La Conchita), respectively). We are grateful with Teodoro Hernández Treviño, Gerardo Arrieta García from the 'Laboratorio Universitario de Geoquímica Isotópica' (LUGIS-UNAM) for their technical support in performing the $^{87}Sr/^{86}Sr$ analyses and to Luis Alberto Aguilar Bautista, Alejandro de León Cuevas, Carlos Sair Flores Bautista, and Jair Garcia Sotelo from the 'Laboratorio Nacional de Visualización Científica Avanzada' (LAVIS/UNAM) who stored our data and provided the computational resources to perform this study. We thank Alejandra Castillo Carbajal and Carina Uribe Díaz for technical support throughout the project.

## Additional information

### Funding

| Funder | Grant reference number | Author |
|---|---|---|
| Welcome Trust Sanger | 208934/Z/17/Z | María C Ávila Arcos |
| PAPIIT-DGAPA-UNAM | IA201219 | María C Ávila Arcos |
| Human Frontier Science Program | RGY0075/2019 | María C Ávila Arcos |

The funders had no role in study design, data collection and interpretation, or the decision to submit the work for publication.

### Author contributions

Axel A Guzmán-Solís, Data curation, Formal analysis, Investigation, Methodology, Visualization, Writing – original draft, Writing – review and editing; Viridiana Villa-Islas, Data curation, Formal analysis, Methodology, Visualization, Writing – review and editing; Miriam J Bravo-López, Data curation, Methodology, Visualization, Writing – review and editing; Marcela Sandoval-Velasco, Conceptualization, Funding acquisition, Resources, Writing – review and editing; Julie K Wesp, Conceptualization, Funding acquisition, Investigation, Resources, Supervision, Writing – review and editing; Jorge A Gómez-Valdés, Resources, Supervision; María de la Luz Moreno-Cabrera, Alejandro Meraz, Resources; Gabriela Solís-Pichardo, Peter Schaaf, Formal analysis, Methodology, Writing – review and editing; Benjamin R TenOever, Resources, Writing – review and editing; Daniel Blanco-Melo, Conceptualization, Data curation, Funding acquisition, Supervision, Visualization, Writing – original draft, Writing – review and editing; María C Ávila Arcos, Conceptualization, Funding acquisition, Investigation, Project administration, Resources, Supervision, Writing – original draft, Writing – review and editing

### Author ORCIDs

Axel A Guzmán-Solís ⬛ http://orcid.org/0000-0001-6878-206X
Jorge A Gómez-Valdés ⬛ http://orcid.org/0000-0001-6996-2732
Benjamin R TenOever ⬛ http://orcid.org/0000-0003-0324-3078
Daniel Blanco-Melo ⬛ http://orcid.org/0000-0002-0256-5019
María C MCAAÁvila Arcos ⬛ http://orcid.org/0000-0003-1691-1696

### Ethics

Human subjects: Permission to process these samples were provided by the INAH Archeology Council with numbers 401.1S.3-2018/1373 and 401.1S.3-2020/1310 for the Hospital San Jose de los Naturales and the Temple of Immaculate Conception (La Conchita), respectively.

### Decision letter and Author response

Decision letter https://doi.org/10.7554/eLife.68612.sa1

Author response https://doi.org/10.7554/eLife.68612.sa2

## Additional files

### Supplementary files

• Supplementary file 1. Viral and phylogenenic analyses. (A) Viral families included in customized capture design. Twenty-five whole genomes from different viruses were included in our customized capture design, as well as a selection of genes from *Herpesviridae* and *Poxviridae* (VARV). Raw or consensus sequences were used as described in Materials and methods. (B). Enrichment yield of HBV-like or B19V-like hits. The metagenomic analysis was carried out with MALT 0.4.0 based on the NCBI RefSeq Viral database, hits were normalized automatically between capture assay(s) and shotgun assay(s) per sample in MEGAN 6.8.0. HSJN177 was considered a negative control for capture based on the shotgun metagenomic analysis. Fold enrichment was calculated as (normalized target capture hits/normalized target shotgun hits). *Two independent NGS libraries were constructed (and then captured) from two different tooth samples belonging to the same individual. NA: not available. (C) . Mapping statistics for ancient viral genome reconstruction. Viral enriched NGS libraries were sequenced and mapped to the human genome (hg19); unmapped reads are shown as total sequences. Number of reads kept in subsequent steps of B19V (COYC4, HSJNC81, HSJN240) and HBV (HSJN194) genome reconstruction are shown as well (Materials and methods). *A second independent round of capture was sequenced deeper in order to obtain a better coverage of the targeted viral ancient genome. **A different tooth was used to construct an independent library for the same individual. #Coverage and depth calculated based on the B19V CDS (AB550331) or whole HBV genome (GQ331046). (C). BEAST analysis of B19V evolutionary rates and MRCA. Overview of B19V Bayesian evolutionary analyses with different molecular clock and priors used. (a) Comparison of median root age and substitution rates. (b) Comparison of MRCA per genotype.

• Supplementary file 2. Human and strontium analyses. (A) Information for the human skeletal remains from which ancient viral genomes were recovered. (B) . $^{87}$Sr/$^{86}$Sr in teeth (enamel) and bones in individuals HSJNC81 and HSJN240. Lix1, Lix2, and Res correspond to the leaching stages. Errors during measurement are presented as one standard deviation with the last two digits (1 sd = ± 1$\sigma_{abs}$). 1 SE(M) = 1 sd/square root n. n = number of runs per analysis. Sr concentrations were determined with the Isotope Dilution technique. ppm = parts per million. The Eimer and Amend (EuA) Sr standard was analyzed during teeth and bone measurements. The certified $^{87}$Sr/ $^{86}$Sr value of this standard is 0.7080 ± 0.0004 (*Fairbairn et al., 1967*).(C) . $^{87}$Sr/$^{86}$Sr sources from West Africa and Trans Mexican Volcanic Belt (TMVB) used for calibration. Compilation of $^{87}$Sr/$^{86}$Sr from West Africa for groups A (*Liégeois et al., 1991*), B (*Bernard-Griffiths et al., 1988*), C (*Peucat et al., 2005*), D (*Ferrière et al., 2010*), E (*Fullgraf et al., 2013*), and F (*Sakyi et al., 2018*) while G (*Solís-Pichardo et al., 2017*) correspond to the TMVB for isotopic comparison.

• Transparent reporting form

• Source data 1. Multi-fasta alignment of all genomes used to perform phylogenetic analysis of B19V.

• Source data 2. Multi-fasta alignment of all genomes used to perform phylogenetic analysis of HBV.

### Data availability

Reconstructed genomes from this study are available in Genbank under accession number MT108214, MT108215, MT108216, MT108217. NGS reads used to reconstruct ancient viral genomes reported in this study are available in Dryad (https://doi.org/10.5061/dryad.5x69p8d2s).

The following dataset was generated:

| Author(s) | Year | Dataset title | Dataset URL | Database and Identifier |
|---|---|---|---|---|
| Avila-Acros MC | 2021 | Data from: Ancient viral genomes reveal introduction of human pathogenic viruses into Mexico during the transatlantic slave trade | https://dx.doi.org/10.5061/dryad.5x69p8d2s | Dryad Digital Repository, 10.5061/dryad.5x69p8d2s |

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
