## [Decision Letter]

**Acceptance summary:**

This work, which characterizes the genetic diversity and spread of ancient viruses circulating in the Americas during the Colonial era, a period marked by the emergence of epidemics in the continent, the cruel slave trade, and the contact between Indigenous and non-Indigenous peoples, will be of interest to microbiologists, anthropologists, and anyone interested in the intersection of history and infectious disease. With appropriate research ethics in place, the authors demonstrate that respectful destructive analyses of ancient DNA from teeth collected from individuals dated to Colonial period Mexico can uncover the source and impact of infectious diseases introduced to the Americas with European colonization.

**Decision letter after peer review:**

Thank you for submitting your article "Ancient viral genomes reveal introduction of HBV and B19V into Mexico during the transatlantic slave trade" for consideration by *eLife*. Your article has been reviewed by 3 peer reviewers, and the evaluation has been overseen by George Perry as the Senior and Reviewing Editor. The following individuals involved in review of your submission have agreed to reveal their identity: C. Eduardo Amorim (Reviewer #1); Charlotte J Houldcroft (Reviewer #2).

Essential Revisions (for the authors):

The reviewers individually and collectively offered praise of your study and its potential for *eLife*. I agree with their assessment. Some excellent reviewer discussions are appended below for your consideration as you revise. There are several points on which expanded detail is required, and other suggestions concerning the balance between the results and discussion. In our consultation session we focused primarily on developing consensus around how to address the question that was consistently raised by the reviewers – what about the bacterial pathogen data? Ultimately, our consensus is that (i) we encourage you to expand on these results as much as possible in the present study. However, (ii) if you prefer to not present the full bacterial results in this study (perhaps due to a forthcoming study with that focus), then at the very least you still need to present an overview of the metagenomic content of the shotgun libraries and address this directly. In addition, considerably further detail into mapping results across all stages of the analysis needs to be provided in order for the reviewers (and future readers) to more fully assess the strength of your results.

*Reviewer #1:*

In my opinion, the major strength of this work is its novelty regarding its questions, results, and ethical imperative of preventing misinterpretation of the results and the stigmatization of the studied subjects/communities. The latter aspect, although only briefly highlighted in the manuscript, is of utmost importance for human population genetics studies in general and for ancient DNA studies in particular, since these studies are reaching the general public beyond academia and research results are often misinterpreted and misused.

Although it is widely known that colonial epidemics had a strong impact on Indigenous populations from the Americas, there is still much speculation on the etiology of these epidemics and direct molecular evidence for the presence of specific viruses is lacking. In this context, Guzmán-Solís et al.'s work pioneer into addressing these questions using state-of-the-art methods and techniques in the genomic sciences. Their results represent the initial steps into characterizing ancient viruses and their spread into the continent during the Colonial era. This manuscript offers a concrete example of how human migrations shaped pathogen dispersal, challenges views about the etiology of colonial epidemics, and adds new evidence for the presence of African viral strains in the Americas. In addition, the phylogenetic contextualization of these ancient viruses sheds new light on the genetic diversity of HBV and B19V in different time transects and the history of the spread of their associated diseases around the world. Authors' claims and conclusions are fully justified by their data and the study is technically sound. I envision Guzmán-Solís et al.'s findings being of interest for scholars from diverse fields which include, human genetics, virology, epidemiology, history of medicine, archaeogenetic, phylogenetics, anthropology, among others.

It is my belief the inclusion of more detailed methods description and explicit research questions would potentially improve this manuscript. For instance, details about how the hits in the metagenomics analysis were normalized are still lacking. In that regard, I am confused with the claim that the "viral abundances were normalized based on the smallest sample size" (line 534). What does "sample size" refer to? Are these sample sizes from the study (N=3 for HSJN; N=1 for COY)? Would that be instead the number of hits? In addition, aspects such as the viral temporal dynamics could be further discussed and the rationale behind this analysis could be more explicit. The manuscript currently does not fully address what it means for the temporal structure to be detected (or not) in their datasets. In addition, I do not see why the root-to-tip distances should depend on sampling times and what is the contribution of this analysis to the understanding of the general questions raised in the paper. It is my belief that other analyses such as population clustering/structure inference would be more appropriate to infer temporal structure. This is not a criticism about the methods employed, but a suggestion for explicitly explaining the rationale behind these analyses and for further discussing the corresponding results. In my detailed review, authors can find additional questions about how double-stranded libraries are expected to perform in the sequencing of single-stranded DNA viruses, as well as whether the overrepresentation of Africans in the human population structure analysis could be biasing their results - however, I believe these potential issues do not invalidate the authors' conclusions in any way. Finally, I miss mapping statistics and contamination estimates for the human ancient genomes.

1. Line 140: regarding the sentence "…contained at least one normalized hit to viral DNA…", could you please specify in the text how this normalization is done (e.g., normalized based on what) and/or what is the purpose of this normalization? This relates to the Methods section, lines 534-536. This part, I believe, would benefit from being more detailed. For instance, I am confused with the claim that the "viral abundances were normalized based on the smallest sample size" (line 534). What does "sample size" refer to? Are these sample sizes from your study (N=3 for HSJN; N=1 for COY)? Would it be instead the number of hits of the sample with the least hits? In addition, how does the number of hits (line 140) translate to viral abundances (line 534)?

2. Line 167: Why only HBV and B19V, if you searched for other viruses as well? Were there cases in which the top hit was not HBV or B19V? If so, why removing these? If there weren't any other cases, then the sentence (line 167) is, in my opinion, somewhat misleading in that it seems a choice was made to focus on HBV and B19V.

3. "DNA extraction and NGS library construction" section: I wonder how your choice for dsDNA libraries could be affecting the detection of B19V, whose genetic material is mostly ssDNA. The cited reference (Meyer and Kircher, 2010) explicitly says that their method allows for preparing libraries from any type of dsDNA and, to my knowledge, they do not mention if their method is adequate for ssDNA. Apparently, this has not been an issue for your experiment, since you were able to detect B19V DNA. Could you explain why?

4. I really appreciate the effort put into designing a capture assay for clinically relevant viruses in human remains and the level of detailing about it in the manuscript. This is a very relevant tool for the community, and I would personally be thrilled to use it to study ancient pathogens in my own research.

5. Virus temporal dynamics/structure (lines 205-223 and Suppl Figure 6): I suggest explaining in the main text what authors mean with "temporal dynamics" or "temporal structure." My understanding is that the root-to-tip distance in phylogenetic trees reflects the number of substitutions in a lineage. Why would this be related to sampling times? I believe you would expect a strong dependency between root-to-tip distances and sampling times if, within each lineage, substitution rates accrued with time and did not depend on other factors. However, at least across mammal lineages, we know that this is not the case because life-history traits, mutation rate modifiers, and effective population sizes are thought to drive variation in the root-to-tip distances across lineages. I wonder if this also applies to viruses. On a related note, even considering this relationship between root-to-tip distances and sampling time to be perfect, I am not sure how it informs about temporal dynamics or temporal structure. Unless this is something well established in the field, I would suggest explaining the rationale behind this analysis. My understanding is that other types of analysis would better characterize the temporal structure. For instance, wouldn't an ADMIXTURE or STRUCTURE analysis be a better choice to characterize temporal structure? Or alternatively, using phylogenetic trees, wouldn't it be better to use the pattern of lineage clustering as an indication for temporal structure (i.e., when lineages from the same time transect cluster together)? Additionally, I strongly suggest the authors further discuss their results in their manuscript. For instance, what does it mean that the HBV does not have a temporal structure while B19V has one? What does it mean that genotype A has a stronger temporal signal than all HBV genotypes together? What does this analysis say about the HBV and B19V lineages found in these human remains in relation to others from the Americas and Africa?

6. I appreciate the paragraph starting at the end of page 15 that explains the introduction of these pathogens is a consequence of the slave trade and not the African peoples themselves. All aDNA studies would benefit from this type of observation that prevents both the misinterpretation of the research results and the stigmatization of the human subjects/communities involved.

*Reviewer #2:*

In this paper, Guzman-Solis and colleagues present a study of the host genetic origins, and selected viral pathogens, of a group of individuals buried in two cemeteries soon after the arrival of European colonisers in Mexico. The paper provides an important insight into the health of people who are often little thought of in conversations about disease brought by European colonisers to the Americas: enslaved and forcibly transported Africans and their descendants. They identify three individuals with parvovirus B19 infections and one person infected with hepatitis B virus. These viruses have been notoriously difficult to accurately date using molecular clock analysis of modern, circulating strains, and thus direct ancient DNA evidence is a very important way of identifying these viral pathogens (which do not leave obvious palaeopathological traces) and dating important events in their evolution and spread.

This paper also presents an example of how to ask questions about infectious disease in the past, and shows the difficulty of trying to identify historical or ancient pathogens even where palaeopathological and written records exist. The authors include much discussion of cocoliztli (what is in the historical record, what theories have been proposed) but this paper is not really an answer to which pathogen(s) caused cocoliztli or whether the individuals studied died of cocoliztli. Ancient DNA studies of infectious diseases are always affected by our current limitations of studying only pathogens with a DNA genome (or a partial DNA stage of their life cycle). Whereas many hypotheses about cocoliztli suggest it was caused by an RNA virus, which would not preserve well in human remains over hundreds of years. The work of the authors points to a view that Europeans - and forcibly transported Africans - brought a package of diseases with them to the Americas. If cocoliztli is in fact a polymicrobial infection or a name for a series of independent epidemics, this will shape how ancient DNA studies of this period of time are conducted in future.

This work selected for individuals who were thought to be of African ancestry (by birth or recent genetic relationships) at one of the sites studied. Therefore it is not surprising that the authors find hepatitis B virus and parvovirus B19 strains which are today associated with African individuals, including in the admixed individual. While it is possible that the forced migration of African individuals to the Americas introduced these viruses to a naïve population, it is also possible that there were (and perhaps still are) local genotypes which have yet to be detected. Patchy modern sampling of these viruses in the Americas today makes the authors' jobs harder. With such a small sample size, partially biased by design towards Africans, this study can't shed much light on whether HBV and B19V were already in the Americas before the arrival of European colonisers or not.

I think this should be published in *eLife*. It's an important contribution to the growing field of ancient viral DNA, and focuses on an exciting period of time and of the world that has been somewhat neglected. I thought the paper had an excellent methods section in particular, very detailed.

The authors don't mention whether bacterial pathogens were also found – I respect that the authors may want to publish such findings elsewhere (if applicable) but I would like the authors to be honest about this – especially as paratyphi has been found associated with remains from C16 Mexico. This is also important to discuss as the authors hint at polymicrobial causes cocoliztli (either within the same host, or circulating at the same time as co/multi-epidemics).

I think the title needs to be slightly changed, as at present it seems to suggest that enslaved Africans brought HBV and B19V to the Americas, which isn't the conclusion of the paper. I thought the discussion of cocoliztli needed focusing, because this paper and its methods aren't really able to answer that question (ie no RNA viruses were considered and there is no mention of malarial parasite DNA or bacteria). However this paper can and very neatly does contribute new knowledge on the milieu of viruses present at the time.

A comment for future experimental design not something to be changed for this study: I thought the design of the viral enrichment panel was sub-optimal. It is missing the rash-causing virus varicella-zoster (chicken pox and shingles) which is highly infectious, associated with more severe disease if primary infection occurs in adulthood, and also thought to be one of the package of diseases carried to the New World by Europeans; KSHV was missing, which would have been common among enslaved Africans and their direct descendants, and can cause endemic disease even in relatively healthy people; adenoviruses would have been another sensible choice given they persist for weeks or months in the host and are spread by respiratory infection. Whereas the choice of cycloviruses perplexed me. I also thought the choice of targets in eg CMV was a bit odd. UL54 (DNA pol) is highly conserved which is good for detection, but wouldn't give you much geographic/phylogenetic signature, and I have only seen functionally significant UL54 mutations after extensive drug treatment. I respect that the composition of the viral enrichment panel will reflect the local interests and experience of the team(s) involved, but if the authors do redesign the panel in future, they should think more broadly among the human DNA viruses and about the sub-genomic regions chosen.

*Reviewer #3:*

Guzmán-Solís, Blanco-Melo, and colleagues present a study of ancient viruses recovered from remains dating to the Colonial period in Mexico. Dental samples from 26 individuals in two different sites with burial contexts suggesting epidemic disease were processed for the recovery of ancient DNA. Following preliminary shotgun results, 12 samples were enriched with a custom bait set designed to capture molecules originating from clinically important viral taxa to examine questions of infectious disease introductions by colonial powers. From these 12 samples, the authors recovered sufficient molecules to reconstruct the genomes of three human parvovirus B19 viruses and one hepatitis B virus.

The weaknesses of this paper are those common to all ancient DNA papers, namely the small sample size, low average coverage, genomes that cannot be fully reconstructed, and the lack of metadata to contextualize epidemiological factors such as acute or chronic infections and cause of death. However, the achievements of this paper far outweigh the weaknesses of the field in general and advance our scientific understanding of colonial impacts on the health and well-being of the Indigenous people of Mexico as well as people of African origin in Mexico during the colonial period. The identification of B19V and HBV within this timeframe does not negate or erase previous scholarship studying the description of colonial epidemics but rather augments those narratives.

All three B19V genomes and the reconstructed HBV genome fall within diversity associated with the African continent suggesting African origins for these infections. African ancestry of the individuals themselves was confirmed via mtDNA for three of the individuals, as well as autosomal data compared to the 1000 Genomes reference panel, and further supported by isotopic analysis of tooth enamel. The fourth individual carried an mtDNA haplogroup associated with Native American populations and autosomal PCA analysis suggested admixed ancestry of Native American and African origin which demonstrates that viruses were transmitted between individuals of different ancestries during this colonial period.

Though the sample size is small, this study demonstrates the effects of European colonization in Mexico through the introduction of humans, both European and enslaved Africans, and pathogenic species carried with these individuals. The interpretation of the data is both biologically sound and conscientiously contextualized for those who had no agency.

This paper was a pleasure to read. I have a few suggestions to improve clarity and transparency, particularly in the supplementary tables, but have no suggestions for additional experimentation or analyses.

– I understand that the emphasis of this paper was to specifically address the presence of viruses in the samples but found it curious that bacteria were not addressed at all. The metagenomic content, even just relative proportions of different Kingdoms, was not conveyed at all. Were no bacterial species identified from the shotgun data? Is there a separate publication expected to cover the bacterial taxa recovered?

– Lines 159-160 and Table 1B. The legend of Table 1B does explain that this is calculated from hits that were normalized by MEGAN. I think that should be reported in lines 159-160 and would also be curious to know if the fold-enrichment values remain approximately constant when not normalized. If reported as ((target capture reads)/(all capture reads)) / ((target shotgun reads)/(all shotgun reads))?

– Lines 207. While Krause-Kyora did perform evolutionary temporal analysis of HBV, their work was preceded by Kahila Bar-Gal et al., 2012 and Patterson Ross et al. 2018. These studies should be cited here as well.

– Line 239. For consistency, swap "shotgun" for "de novo".

– Lines 468-470. While not as old as the samples reported in this study and by Barquera et al., and also not from archaeological contexts, the genomes reported in both Schrick et al., 2017 and Duggan, Klunk et al., 2020 are ancient viral genomes from the Americas. The authors may wish to rephrase this passage.

– Lines 520-524. Missing from this discussion and any of the supplementary tables is reference to how deeply the libraries were shotgun sequenced. I understand the sensitivity in releasing human genomic data, particularly from contexts such as these, and have no objection to only the viral reads appearing on Dryad, I do believe that the scope of all data generated needs to be represented somewhere in the paper. There is also no indication anywhere of the number of reads that were used to generate the mitochondrial genomes or the autosomal data for the PCA plots which takes away some of the reader's ability to evaluate the findings. It also makes it difficult to interpret some of the viral data, such as Figure 1c.

– Figure 2c. Adding a label as part of the plot to indicate HBV as the reference would be beneficial.

– Throughout. The nomenclature and meaning of DS1 and DS2 is challenging to follow as both HBV and B19V seem to have DS1s and DS2s but Table 1c lists only DS1 without specifying if this a singular dataset, separate datasets for HBV and B19V applied to separate samples in the same column, or two datasets merged together.

---

## [Author Response]

Reviewer #1:1. Line 140: regarding the sentence "…contained at least one normalized hit to viral DNA…", could you please specify in the text how this normalization is done (e.g., normalized based on what) and/or what is the purpose of this normalization? This relates to the Methods section, lines 534-536. This part, I believe, would benefit from being more detailed. For instance, I am confused with the claim that the "viral abundances were normalized based on the smallest sample size" (line 534). What does "sample size" refer to? Are these sample sizes from your study (N=3 for HSJN; N=1 for COY)? Would it be instead the number of hits of the sample with the least hits? In addition, how does the number of hits (line 140) translate to viral abundances (line 534)?

We appreciate the reviewer’s concerns. To avoid confusion, we now provide a more detailed description about our motivation to normalize the hits in the main text (lines 130-133); and the way MEGAN does it at methods (lines 555-559). Just to clarify, throughout the manuscript we refer to the number of hits when talking about a specific virus or viral family, while we use viral abundances in a more general way, when referring to a group of viruses.

Lines 130-133:

“… revealed seventeen samples containing at least one normalized hit to viral DNA (abundances were normalized to the smallest library, since each sample had different number of reads) (Methods)”

Lines 555-559:

“The RMA files were used for the normalization of the viral abundances based on the library with the smallest number of reads (default, (count of class/total count of sample)* count of smallest sample) and compared to all the samples from the same archeological site with MEGAN 6.8.0”

2. Line 167: Why only HBV and B19V, if you searched for other viruses as well? Were there cases in which the top hit was not HBV or B19V? If so, why removing these? If there weren't any other cases, then the sentence (line 167) is, in my opinion, somewhat misleading in that it seems a choice was made to focus on HBV and B19V.

We thank the reviewer for pointing out this source of confusion. Upon enrichment, only four libraries showed an increase in virus-like hits corresponding to the Hepadnaviridae and Parvoviridae families, therefore we focused our attention on these viral families (lines 154-158). Additionally, to double check that the reads were actually from HBV or B19V, we took the reads mapped to HBV or B19V and then queried these against the nr database. If the top hit was to HBV or B19V, then we retained the reads, otherwise we excluded it before assembling the genome. We explain this now with more detail in lines 170-176 and methods (lines 663-665 and 675-677).

Lines 154-158:

“Only one post-capture library had a ~100-fold increase of Hepadnaviridae-like hits (HBV), while three more libraries had a ~50-200-fold increase of Parvoviridae-like hits (B19V) (Figure 1c, Supplementary file 1B), compared to their corresponding pre-capture libraries (Methods).”

Lines 170-176:

“We verified the authenticity of the reads mapped to HBV (BWA) or B19V (BWA/blastn) in the enriched libraries (Methods) by querying the reads against the non-redundant (nr) NCBI database using megaBLAST (Altschul et al., 1990). This step was performed to avoid including in the genome assembly reads that were mapped by BWA or blastn as HBV or B19V, but with a similar identity to a different taxon in the nr database (and absent in DS1. Methods). Therefore, we only retained reads for which the top hit was to either B19V or HBV (Supplementary file 1C).”

Lines 663-665:

“All the HBV mapped reads were analyzed through megaBLAST (Altschul et al., 1990) using the whole NCBI nr database to verify they were assigned uniquely to HBV (carried out with Krona 2.7 (Ondov et al., 2011)).”

Lines 675-677:

“The resulting reads were analyzed with megaBLAST (Altschul et al., 1990) using the whole NCBI nr database to verify the top hit was to B19V (carried out with Krona 2.7 (Ondov et al., 2011)).”

3. "DNA extraction and NGS library construction" section: I wonder how your choice for dsDNA libraries could be affecting the detection of B19V, whose genetic material is mostly ssDNA. The cited reference (Meyer and Kircher, 2010) explicitly says that their method allows for preparing libraries from any type of dsDNA and, to my knowledge, they do not mention if their method is adequate for ssDNA. Apparently, this has not been an issue for your experiment, since you were able to detect B19V DNA. Could you explain why?

We acknowledge the reviewer's concerns as it makes us realize that we weren’t clear in our attempt to discuss this in the first version of the manuscript. We now add a more thorough explanation in lines 304-318.

Lines 304-318:

“The recovery of aDNA from B19V, which has a ssDNA genome (with dsDNA terminal repeats), in previous studies (Mühlemann, Margaryan, et al., 2018) as well as in our samples, is noteworthy considering the NGS libraries were constructed using dsDNA as a template. Therefore, we would not expect to recover the ssDNA from B19V with this library building method. However, it is known that dsDNA intermediates form during the B19V replication cycle (Ganaie and Qiu, 2018), and that throughout the viral infection the replicating genomes are present in both the ssDNA and dsDNA forms. The sequences we retrieved must therefore correspond to the cell-free dsDNA replication intermediates. This is coherent with the peculiar coverage pattern on the B19V genome, where the dsDNA hairpins at its terminal sites and are highly covered, reflecting a better stability of these regions over time (Figure 2b). Similarly, the partially circular dsDNA genome from HBV was poorly covered at the ssDNA region (Figure 2c), which also goes through a dsDNA phase during replication, a similar coverage is reported in three previous ancient HBV genomes (Krause-Kyora et al., 2018)”

4. I really appreciate the effort put into designing a capture assay for clinically relevant viruses in human remains and the level of detailing about it in the manuscript. This is a very relevant tool for the community, and I would personally be thrilled to use it to study ancient pathogens in my own research.

We thank the reviewer for the interest. We have included all the details to reproduce the probe set in the manuscript. However we can also share the sequences for each probe if it would be useful for others.

5. Virus temporal dynamics/structure (lines 205-223 and Suppl Figure 6): I suggest explaining in the main text what authors mean with "temporal dynamics" or "temporal structure." My understanding is that the root-to-tip distance in phylogenetic trees reflects the number of substitutions in a lineage. Why would this be related to sampling times? I believe you would expect a strong dependency between root-to-tip distances and sampling times if, within each lineage, substitution rates accrued with time and did not depend on other factors. However, at least across mammal lineages, we know that this is not the case because life-history traits, mutation rate modifiers, and effective population sizes are thought to drive variation in the root-to-tip distances across lineages. I wonder if this also applies to viruses. On a related note, even considering this relationship between root-to-tip distances and sampling time to be perfect, I am not sure how it informs about temporal dynamics or temporal structure. Unless this is something well established in the field, I would suggest explaining the rationale behind this analysis. My understanding is that other types of analysis would better characterize the temporal structure. For instance, wouldn't an ADMIXTURE or STRUCTURE analysis be a better choice to characterize temporal structure? Or alternatively, using phylogenetic trees, wouldn't it be better to use the pattern of lineage clustering as an indication for temporal structure (i.e., when lineages from the same time transect cluster together)? Additionally, I strongly suggest the authors further discuss their results in their manuscript. For instance, what does it mean that the HBV does not have a temporal structure while B19V has one? What does it mean that genotype A has a stronger temporal signal than all HBV genotypes together? What does this analysis say about the HBV and B19V lineages found in these human remains in relation to others from the Americas and Africa?

We thank the reviewer for the constructive suggestions. We realized our analyses required further explanation to avoid confusion and a better discussion. We have modified the manuscript to explain the rationale of the analysis in lines 215-222:

Lines 215-222:

“In order to use our reconstructed viral genomes as molecular fossils to recalibrate each virus phylogeny and perform evolutionary inferences; we first needed to estimate if the phylogenetic relationships among B19V or HBV genomes had a temporal structure (i.e. sufficient genetic change between sampling times to reconstruct a statistical relationship between genetic divergence and time) (Rambaut et al., 2016). In the context of viruses, temporal structure is canonically tested with a root-to-tip distance anxd date-randomization analyses (see Firth et al., 2010; Rieux and Balloux, 2016).”

We do not believe ADMIXTURE or STRUCTURE approaches assess in a quantitative way (as we do with the used tests) the presence or absence of temporal structure.

6. I appreciate the paragraph starting at the end of page 15 that explains the introduction of these pathogens is a consequence of the slave trade and not the African peoples themselves. All aDNA studies would benefit from this type of observation that prevents both the misinterpretation of the research results and the stigmatization of the human subjects/communities involved.

We thank the reviewer for the comment. This is a very delicate and painful topic and we did our best to try to address it with the most sensibility to avoid misinterpretation and further damage.

Reviewer #2:I think this should be published in eLife. It's an important contribution to the growing field of ancient viral DNA, and focuses on an exciting period of time and of the world that has been somewhat neglected. I thought the paper had an excellent methods section in particular, very detailed.

We appreciate the encouraging comments by the reviewer.

The authors don't mention whether bacterial pathogens were also found – I respect that the authors may want to publish such findings elsewhere (if applicable) but I would like the authors to be honest about this – especially as paratyphi has been found associated with remains from C16 Mexico. This is also important to discuss as the authors hint at polymicrobial causes cocoliztli (either within the same host, or circulating at the same time as co/multi-epidemics).

As assumed by the reviewer, the bacterial pathogens in the collections analyzed here were described in a different publication that we now refer to (Bravo-Lopez et al., 2020). In that paper, we reported the identification of oral pathogens in several ancient individuals, including two individuals (HSJN194 and HSJN204) from this study. We did not identify other pathogenic bacteria in the individuals from this study. Nonetheless, in the spirit of transparency we now include more comprehensive metagenomic profiles for the individuals in this study as figure supplement (Figure 1—figure supplement 4-7). We also include a brief description of the taxonomic results in lines 161-168 and the relevant information in the Methods section (line 560-566).

Lines 161-168.

“Independently, a metagenomic analysis using Kraken2 (Wood et al., 2019) and Pavian (Breitwieser and Salzberg, 2020) was performed on the non-human (unmapped) reads as part of a different study (Bravo-Lopez et al., 2020). Our samples presented bacterial constituents of human oral and soil microbiota at different proportions between the samples (Figure 1—figure supplement 4-7). Although no lethal bacterial pathogen was retrieved, some ancient dental pathogens (Tannerella forsythia) were reconstructed and described in more detail by Bravo-Lopez et al., 2020 (Figure 1—figure supplement 4-7)”

Lines 560-566.

“Independently, unmapped reads (non-human) were taxonomically classified with Kraken2 (Wood et al., 2019) using a reference database composed of NCBI RefSeq bacterial, archaea and viral genomes (downloaded on November 3rd, 2017). The Kraken2 output was transformed to a BIOM-format table using Kraken-biom (https://github.com/smdabdoub/kraken-biom) and then visualized with Pavian (Breitwieser and Salzberg, 2020). Detailed description of the results can be found in Bravo-Lopez et al., 2020.”

I think the title needs to be slightly changed, as at present it seems to suggest that enslaved Africans brought HBV and B19V to the Americas, which isn't the conclusion of the paper. I thought the discussion of cocoliztli needed focusing, because this paper and its methods aren't really able to answer that question (ie no RNA viruses were considered and there is no mention of malarial parasite DNA or bacteria). However this paper can and very neatly does contribute new knowledge on the milieu of viruses present at the time.

Following the suggestion by another reviewer we now propose an alternative title:

“Ancient viral genomes reveal introduction of human pathogenic viruses into Mexico during the transatlantic slave trade”

Additionally, we acknowledge that our study is really inconclusive regarding the causative agent of Cocoliztli, and that the emphasis on this can be misleading. We have therefore removed a paragraph in the introduction detailing the symptoms and current hypotheses of the causative agents. Furthermore we have made substantial changes in this regard in the discussion, also removing emphasis on Cocoliztli. With these changes, we believe the introduction and discussion are more focused and do not hint to the study being in any way able to conclude anything about Cocoliztli, but highlight the overall relevance of learning about the viruses circulating in Colonial Mexico.

A comment for future experimental design not something to be changed for this study: I thought the design of the viral enrichment panel was sub-optimal. It is missing the rash-causing virus varicella-zoster (chicken pox and shingles) which is highly infectious, associated with more severe disease if primary infection occurs in adulthood, and also thought to be one of the package of diseases carried to the New World by Europeans; KSHV was missing, which would have been common among enslaved Africans and their direct descendants, and can cause endemic disease even in relatively healthy people; adenoviruses would have been another sensible choice given they persist for weeks or months in the host and are spread by respiratory infection. Whereas the choice of cycloviruses perplexed me. I also thought the choice of targets in eg CMV was a bit odd. UL54 (DNA pol) is highly conserved which is good for detection, but wouldn't give you much geographic/phylogenetic signature, and I have only seen functionally significant UL54 mutations after extensive drug treatment. I respect that the composition of the viral enrichment panel will reflect the local interests and experience of the team(s) involved, but if the authors do redesign the panel in future, they should think more broadly among the human DNA viruses and about the sub-genomic regions chosen.

The choices made during the design of the probes had the objective to maximize the number of viral families (and subfamilies) susceptible to be captured by the probes, while having a strict length constraint given the limits of the kit size that we were able to afford at the time. Given the large genome size of the Herpesviridae family, we choose a few genes of only one representative virus from each of the three Herpesvirus subfamilies (Α: HSV1, Β: CMV and Γ: EBV), based on their prevalence in the modern population. Nevertheless, we appreciate and acknowledge the reviewer’s concerns and will take them into account for future more comprehensive probe designs. We have clearly stated the size constraints of the probe kit in methods (line 149-150).

Line 149-150:

“(Given the size constraints of the probe kit, only a couple of genes were selected from some viral families Methods, Supplementary file 1A).”

Reviewer #3:This paper was a pleasure to read. I have a few suggestions to improve clarity and transparency, particularly in the supplementary tables, but have no suggestions for additional experimentation or analyses.– I understand that the emphasis of this paper was to specifically address the presence of viruses in the samples but found it curious that bacteria were not addressed at all. The metagenomic content, even just relative proportions of different Kingdoms, was not conveyed at all. Were no bacterial species identified from the shotgun data? Is there a separate publication expected to cover the bacterial taxa recovered?

As stated in a previous response to reviewer 2, we screened for bacterial aDNA as part of a different study. We now include the reference to said study and a broader metagenomic profile for the individuals analyzed here (Figure 1—figure supplement 4-7). See response to reviewer 2.

– Lines 159-160 and Table 1B. The legend of Table 1B does explain that this is calculated from hits that were normalized by MEGAN. I think that should be reported in lines 159-160 and would also be curious to know if the fold-enrichment values remain approximately constant when not normalized. If reported as ((target capture reads)/(all capture reads)) / ((target shotgun reads)/(all shotgun reads))?

The fold-enrichment reported at Supplementary File 1B was calculated as (normalized target capture hits / normalized target shotgun hits) as now mentioned at figure legend (Line 1394) and lines 154-158.

The normalization between shotgun and capture was performed because each sequencing round had a different number of reads so we would expect a higher number of reads in the library sequenced deeper. The reported fold enrichment in Supplementary File 1B is based on the library with fewer reads. We now indicate the way MEGAN normalizes data and our motivation to do this step at lines 130-133 and 555-559.

Lines 154-158:

“Only one post-capture library had a ~100-fold increase of Hepadnaviridae-like hits (HBV), while three more libraries had a ~50-200-fold increase of Parvoviridae-like hits (B19V) (Figure 1c, Supplementary file 1B), compared to their corresponding pre-capture libraries (Methods)”

Lines 130-133:

“… revealed seventeen samples containing at least one normalized hit to viral DNA (abundances were normalized to the smallest library, since each sample had different number of reads) (Methods)”

Lines 555-559:

“The RMA files were used for the normalization of the viral abundances based on the library with the smallest number of reads (default, (count of class/total count of sample)* count of smallest sample) and compared to all the samples from the same archeological site with MEGAN 6.8.0”

– Lines 207. While Krause-Kyora did perform evolutionary temporal analysis of HBV, their work was preceded by Kahila Bar-Gal et al., 2012 and Patterson Ross et al., 2018. These studies should be cited here as well.

We thank the reviewer for pointing out this omission. The citation for Patterson Ross et al., 2018 has now been included in the manuscript at line 222. KahilaBar-Gal et al., 2012 performed their molecular-clock analysis directly in beast without corroborating the presence of temporal structure in their dataset, so we did not include the citation.

Lines 222:

“Similarly to previous studies(Krause-Kyora et al., 2018; Patterson Ross et al., 2018)…”

– Line 239. For consistency, swap "shotgun" for "de novo".

Modified at line 254.

– Lines 468-470. While not as old as the samples reported in this study and by Barquera et al., and also not from archaeological contexts, the genomes reported in both Schrick et al., 2017 and Duggan, Klunk et al., 2020 are ancient viral genomes from the Americas. The authors may wish to rephrase this passage.

We appreciate the suggestion and have now included the suggested citations and modified the text accordingly.

Lines 489-492:

“The isolation and characterization of these ancient HBV and B19V genomes represent an important contribution to the ancient viral genomes reported in the Americas (Barquera et al., 2020; Duggan et al., 2020; Schrick et al., 2017).”

– Lines 520-524. Missing from this discussion and any of the supplementary tables is reference to how deeply the libraries were shotgun sequenced. I understand the sensitivity in releasing human genomic data, particularly from contexts such as these, and have no objection to only the viral reads appearing on Dryad, I do believe that the scope of all data generated needs to be represented somewhere in the paper.

Thank you for the suggestion, the shotgun statistics and human analyses (somatic and mitochondrial) for the individuals reported in this study are now included at Supplementary File 2A.

There is also no indication anywhere of the number of reads that were used to generate the mitochondrial genomes or the autosomal data for the PCA plots which takes away some of the reader's ability to evaluate the findings. It also makes it difficult to interpret some of the viral data, such as Figure 1c.

We thank the reviewer for pointing out this omission. The human mapping statistics including the number of SNPs used for the PCA and mitochondrial analyses is now included at Supplementary File 2A.

– Figure 2c. Adding a label as part of the plot to indicate HBV as the reference would be beneficial.

We appreciate this suggestion and have modified Figure 2c accordingly.

– Throughout. The nomenclature and meaning of DS1 and DS2 is challenging to follow as both HBV and B19V seem to have DS1s and DS2s but Table 1c lists only DS1 without specifying if this a singular dataset, separate datasets for HBV and B19V applied to separate samples in the same column, or two datasets merged together.

We thank the reviewer for this comment. We have now modified the nomenclature of the distinct datasets as follow:

HBV_DS1

HBV_DS2

B19V_DS1

B19V_DS2